

# Collection/aggregation algorithms in Lagrangian cloud microphysical models: Rigorous evaluation in box model simulations

Simon Unterstrasser[1], Fabian Hoffmann[2], and Marion Lerch[1]

[1]Deutsches Zentrum für Luft- und Raumfahrt (DLR) – Institut für Physik der Atmosphäre, Oberpfaffenhofen, 82234 Wessling, Germany.
[2]Leibniz Universität Hannover – Institute of Meteorology and Climatology, 30419 Hannover, Germany.

*Correspondence to:* Simon Unterstrasser: simon.unterstrasser@dlr.de

**Abstract.** Recently, several Lagrangian microphysical models have been developed which use a
large number of (computational) particles to represent a cloud. In particular, the collision process
leading to coalescence of cloud droplets or aggregation of ice crystals is implemented differently
in the various models. Three existing implementations are reviewed and extended, and their perfor-
mance is evaluated by a comparison with well established analytical and bin model solutions. In this
first step of rigorous evaluation, box model simulations with collection/aggregation being the only
process considered have been performed for the three well-known kernels of Golovin, Long and
Hall.
Besides numerical parameters like the time step and the number of simulation particles (SIPs)
used, the details of how the initial SIP ensemble is created from a prescribed analytically defined
size distribution is crucial for the performance of the algorithms. Using a constant weight tech-
nique as done in previous studies greatly underestimates the quality of the algorithms. Using better
initialisation techniques considerably reduces the number of required SIPs to obtain realistic re-
sults. From the box model results recommendations for the collection/aggregation implementation
in higher dimensional model setups are derived. Suitable algorithms are equally relevant to treating
the warm-rain process and aggregation in cirrus.

## 1 Introduction

The collection of cloud droplets or the aggregation of ice crystals is an important process in liquid
and ice clouds. By changing the size, number, and in the case of ice the shape of hydrometeors,
collection and aggregation affect the microphysical behaviour of clouds and thereby their role in the
climate system.
The warm rain process (i.e. the production of precipitation in clouds in the absence of ice) de-
pends essentially on the collision and subsequent coalescence of cloud droplets. At its initial stage,





however, condensational growth governs the activation of aerosols and the following growth of cloud
droplets, which might initiate the collection process if they become sufficiently large. Then, collec-
tion produces drizzle or raindrops, which are able to precipitate from the cloud, affecting lifetime
and organisation of clouds (e.g. Albrecht, 1989; Xue et al., 2008).
In ice clouds, sedimentation, deposition growth and in particular radiative properties depend on
the ice crystals' habits (Sölch and Kärcher, 2011, and references therein). Ice aggregates scatter
more strongly shortwave radiation than pure ice crystals of the same mass. Recent simulation results
suggest that contrail-cirrus and natural cirrus can be strongly interwoven. In the mixing area with
ice crystals of both origins being present, a prominent bimodal spectrum occurs and enhances the
probability of collisions (Unterstrasser et al., 2016).
The temporal change of the droplet number distribution by the collision and subsequent coales-
cence of droplets (or any other particles) is described by the stochastic collection equation (SCE),
also known as kinetic collection equation, coagulation equation, Smoluchowksi or population bal-
ance equation (e.g. Wang et al., 2007). It yields:

$$\frac{\partial f_m(m,t)}{\partial t} = \frac{1}{2} \int_0^m K(m',m-m') f_m(m',t) f_m(m-m',t)\, dm'$$

$$- \int_0^\infty K(m,m') f_m(m,t) f_m(m',t)\, dm', \tag{1}$$

where $f_m(m)dm$ is the number concentration within an infinitesimal interval around the mass $m$.
The first term (gain term) accounts for the coalescence of two smaller droplets forming a new
droplet with mass $m$, the second term (loss term) accounts for the coalescence of $m$-droplets with
any other droplets forming a larger droplet. The collection kernel $K(m,m')$ describes the rate
by which a $m$-droplet-$m'$-droplet-collection occurs. Due to the symmetry of the collection kernel
$(K(m,m') = K(m',m))$ the first term of the right-hand side can also be written as $\int_0^{m/2} K(m',m-$
$m') f_m(m',t) f_m(m-m',t)\, dm'$.
For several kernel functions (mostly of polynomial form) analytic solutions exist for specific initial
distributions (Golovin, 1963; Berry, 1967; Scott, 1968). The Golovin kernel (sum of masses) is given
by
$$K(m,m') = b\,(m+m'). \tag{2}$$
Solutions for more realistic kernels (Long, 1974; Hall, 1980; Wang et al., 2006) and arbitrary initial
distribution can be obtained with various numerical methods mainly using a bin representation of the
droplet size distribution (Berry and Reinhardt, 1974; Tzivion et al., 1987; Bott, 1998; Simmel et al.,
2002; Wang et al., 2007). The hydrodynamic kernel is defined as
$$K(r,r') = \pi(r+r')^2\, |w_{sed}(r) - w_{sed}(r')|\, E_c(r,r'), \tag{3}$$





based on the radius $r$ and the sedimentation velocity $w_{sed}$. Parametrisations of the collection ef-
ficiency $E_c$ are given, e.g. by Long (1974) or Hall (1980). In the above formula, the differen-
tial sedimentation is the driver of collections. No same-size collisions can occur, i.e. $K(r,r) = 0$.
More sophisticated expressions for $K(r,r')$ have been derived to include turbulence enhancement
of the collisional growth, which also allow same-size collisions ($K(r,r) > 0$) (e.g. Ayala et al., 2008;
Grabowski and Wang, 2013; Chen et al., 2016).
Solving (1) demands simplifications in the representation of the droplet spectrum for which sev-
eral numerical models have been developed. Spectral-bin models (e.g. Khain et al., 2000) represent
the spectrum by dividing it into several intervals, so-called bins. This approach enables the predic-
tion of the temporal development of the droplet number concentration in each bin by using finite
differences or more sophisticated numerical techniques (e.g. Bott, 1998). The accuracy of these
models is primarily determined by the number of used bins (usually on the order of 100), which
makes them computationally challenging and prohibits their use in day-to-day applications like nu-
merical weather prediction. Less challenging but less accurate, cloud microphysical bulk models
compute the temporal change of integral quantities of the droplet spectrum (e.g. Kessler, 1969;
Khairoutdinov and Kogan, 2000; Seifert and Beheng, 2001). These are usually equations for the
temporal evolution of bulk mass (so-called one-moment schemes), and additionally number con-
centration (two-moment schemes) or radar reflectivity (three-moment schemes), which describe the
change of the entities of cloud droplets and rain drops (in the case of warm clouds). The separation
radius between cloud droplets and rain drops depends on the details of the bulk scheme, but generally
cloud droplets (up to 20 to $40\,\mu$m in radius) are assumed to have negligible sedimentation fall veloci-
ties, while larger drops, frequently subsumed as rain drops, have significant sedimentation velocities
to cause collision/coalescence. The interactions of cloud and rain drops are therefore described in
terms of self-collection (coalescence of cloud (rain) drops resulting in cloud (rain) drops), autocon-
version (coalescence of cloud droplets resulting in rain drops) and accretion (collection of cloud
droplets by rain drops). A third alternative for computing cloud microphysics has been developed
in the recent years: Lagrangian cloud models (LCMs). These models represent cloud microphysics
on the basis of individual particles. Similar to spectral-bin models, LCMs enable the detailed rep-
resentation of droplet spectra but inherently avoid spurious numerical diffusion in condensational
and collisional growth usually affecting the results of spectral-bin models (Andrejczuk et al., 2010;
Arabas and Shima, 2013).
To our knowledge, five fully coupled LCMs for warm clouds exist, which are described in Andrejczuk et al.
(2008); Shima et al. (2009); Riechelmann et al. (2012); Arabas et al. (2015) and Naumann and Seifert
(2015) and have been extended or applied in various problems (e.g. Andrejczuk et al., 2010; Arabas and Shima,
2013; Lee et al., 2014; Hoffmann et al., 2015). For ice clouds, three models exist (Paoli et al., 2004;
Shirgaonkar and Lele, 2006; Sölch and Kärcher, 2010) which have been applied to natural cirrus
(Sölch and Kärcher, 2011) and, in particular, to contrails (e. g. Paoli et al., 2013; Unterstrasser, 2014;



Unterstrasser and Görsch, 2014). In the context of ice clouds and warm clouds, different names
are used for processes that are similar, in particular in terms of their numerical treatment (depo-
sition/sublimation vs. condensation/evaporation, collection vs. aggregation). Conceptually similar
are particle based approaches in aerosol physics (Riemer et al., 2009; Maisels et al., 2004) which
account for coagulation of aerosols (DeVille et al., 2011; Kolodko and Sabelfeld, 2003).
So far, no consistent terminology has been used in the latter publications. Various names have
been used for the same things by various authors. We point out that super droplet, computational
droplet and simulation particle (SIP) all have the same meaning and refer to a bunch of identical
real cloud droplets (or ice crystals). The number of real droplets represented in a SIP is denoted
as weighting factor or multiplicity. Moreover, Lagrangian approaches in cloud physics have been
named Lagrangian Cloud Model (LCM), super droplet method (SDM) or particle based method. In
this paper, we use the terms SIP, weighting factor $\nu_{sim}$ and LCM. Here droplet refers to either real
droplets or ice crystals.
Usually, only the liquid water or the ice of a cloud are described with a Lagrangian representation,
whereas all other physical quantities (like velocity, temperature and water vapour concentration) are
described in Eulerian space (see also discussion in Hoffmann, 2016). SIPs have discrete positions
$\mathbf{x}_p = (x_p, y_p, z_p)$ within a grid box. The position is regularly updated obeying the transport equation
$\partial \mathbf{x}_p / \partial t = \mathbf{u}$. Microphysical processes like sedimentation and droplet growth are treated individually
for each SIP. Interpolation methods can be used to evaluate the Eulerian fields at the specific SIP
positions. This implicitly assumes that all $\nu_{sim}$ droplets of the SIPs are located at the same position.
On the other hand, the droplets of a SIP are assumed to be well-mixed in the grid box in LCM
treatment of collection and sometimes condensation. Then, the number concentration represented
by a single SIP, e. g., is given by $\nu_{sim}/\Delta V$, where $\Delta V$ is the volume of the grid box.
Lists of used symbols and abbreviation are given in Tables 1 and 2.
## 2   Description of the various collection/aggregation implementations
We use the terminology of Berry (1967), where $f_{\ln r}$ and $g_{\ln r}$ denote the number and mass density
function with respect to the logarithm of droplet radius $\ln r$. The relations $g_{\ln r}(r) = m f_{\ln r}(r)$ and
$f_{\ln r}(r) = 3m f_m(m)$ hold. The latter designates the number density function with respect to mass
and obeys the transformation property of distributions: $f_y(y)dy = f_x(x(y))dx$. For consistency with
previous studies, $g_{\ln r}$ is used for plotting purposes, whereas $f_m$ and $g_m$ are more relevant in the
following analytical derivations.
The moments of order $k$ of the mass distribution $f_m$ (= number density function with respect to
mass) are defined as:
$$\lambda_k(t) = \int m^k f_m(m,t)dm. \qquad (4)$$





**Table 1.** List of symbols.

| Symbol | Value/Unit | Meaning |
| --- | --- | --- |
| $f_m, \tilde{f}_m$ | $\mathrm{kg}^{-1}\,\mathrm{m}^{-3}$, 1 | (normalised) droplet number concentration per mass interval |
| $g_m, g_{\ln r}$ | $\mathrm{m}^{-3}$, $\mathrm{kg}\,\mathrm{m}^{-3}$ | droplet mass concentration per mass interval/per logarithmic radius interval |
| $m, m'$ | kg | mass of a single real droplet |
| $m_{bb}$ | kg | bin boundaries of the bin grid |
| $\bar{m} = \lambda_1/\lambda_0 = \mathcal{N}/\mathcal{M}$ | kg | mean mass of all droplets |
| $n_{bin,l}$ | 1 | droplet number in bin $l$ |
| $r, r'$ | m | droplet radius |
| $r_{lb}$ | m | threshold radius in $\nu_{random,lb}$-init |
| $r_{critmin}$ | m | lower cut-off radius in singleSIP-init |
| $w_{sed}$ | $\mathrm{m}\,\mathrm{s}^{-1}$ | sedimentation velocity |
| $DNC = \lambda_0$ | $\mathrm{m}^{-3}$ | droplet number concentration |
| $E_c$ | 1 | collection/aggregation efficiency |
| $K$ | $\mathrm{m}^3\,\mathrm{s}^{-1}$ | collection/aggregation kernel |
| $LWC = \lambda_1$ | $\mathrm{kg}\,\mathrm{m}^{-3}$ | droplet mass concentration, liquid water content |
| $M_{bin,l}$ | kg | total droplet mass in bin $l$ |
| $N_{SIP}$ | 1 | number of SIPs |
| $N_{BIN}$ | 1 | number of bins |
| $\alpha_{low}, \alpha_{med}, \alpha_{high}$ | 1 | parameters of the $\nu_{random}$-init method. |
| $\Delta t$ | s | time step |
| $\Delta V$ | $\mathrm{m}^3$ | grid box volume |
| $\eta$ | 1 | parameter in RMA algorithm and singleSIP-init method |
| $\kappa$ | 1 | number of bins per mass decade |
| $\lambda_k$ | $\mathrm{kg}^k\,\mathrm{m}^{-3}$ | moments of the order k |
| $\mu$ | kg | single droplet mass of a SIP |
| $\nu_{critmax}$ | 1 | maximum number of droplets represented by a SIP |
| $\nu_{critmin}$ | 1 | minimum number of droplets represented by a SIP |
| $\nu$ | 1 | number of droplets represented by a SIP |
| $\xi$ | 1 | splitting parameter of AON algorithm |
| $\chi = \mu\,\nu, \tilde{\chi} = \chi/\mathcal{M}$ | kg, 1 | total droplet mass of a SIP |
| $\mathcal{N} = \lambda_0 \Delta V$ | 1 | total droplet number |
| $\mathcal{M} = \lambda_1 \Delta V$ | kg | total droplet mass |
| $\mathcal{Z} = \lambda_2\,\Delta V$ | $\mathrm{kg}^2$ | second moment of droplet mass distribution (radar reflectivity) |





**Table 2.** List of abbreviations.

| | | | |
|---|---|---|---|
| AON | All-Or-Nothing algorithm | AIM | Average Impact algorithm |
| DSD | droplet size distribution | LCM | Lagrangian Cloud Model |
| PDF | probability density function | RMA | Remapping algorithm |
| SIP | simulation particle | | |

The low order moments represent the number concentration ($DNC = \lambda_0$) and the mass concentra-
tion ($LWC = \lambda_1$). The analogous extensive properties $\lambda_k(t)\,\Delta V$ are the total droplet number $\mathcal{N}$,
total droplet mass $\mathcal{M}$ and radar reflectivity ($\mathcal{Z} = \lambda_2\,\Delta V$). For a given SIP ensemble, the moments
can be simply computed by
$$\lambda_{k,SIP}(t) = \left(\sum_{i=0}^{N_{SIP}} \nu_i \mu_i{}^k\right) / \Delta V, \tag{5}$$
where $\mu_i$ is the single droplet mass of SIP $i$ and $N_{SIP}$ is the number of SIPs inside a grid box. For
reasons of consistency with Wang et al. (2007), we translate the SIP ensemble into a mass distribu-
tion $g_m$ in bin representation and then compute the moments with the formula
$$\lambda_{k,BIN}(t) = \sum_{i=0}^{N_{BIN}} g_m(m_i,t)(\tilde{m}_{bb,l})^{k-1}\frac{\ln 10}{3\,\kappa} \tag{6}$$
(cf. with their equation 48).
The initialisation is successful for a given parameter set, if the moments of the SIP ensemble
$\lambda_{k,SIP}$ are close to the analytical values $\lambda_{k,anal}$. For an exponential distribution (as used in this
study), the probability density function (PDF) reads as
$$f_m(m) = \frac{\mathcal{N}}{\Delta V \bar{m}} \exp\left(-\frac{m}{\bar{m}}\right), \tag{7}$$
the moments are given analytically by
$$\lambda_{k,anal}(t) = (k-1)!\,\mathcal{N}\,\bar{m}^k/\Delta V, \tag{8}$$
where $k!$ is the faculty of $k$ and $\bar{m} = \mathcal{M}/\mathcal{N}$ the mean mass (Rade and Westergren, 2000).
Throughout this study, the initial parameters of the droplet size distribution (DSD) are $DNC_0 =$
$2.97 \times 10^8$ m$^{-3}$ and $LWC_0 = 10^{-3}$ kg m$^{-3}$ (implying a mean radius of $9.3\,\mu$m) as in Wang et al.
(2007). The higher moments are $\lambda_{2,anal} = 6.74 \times 10^{-15}$ kg$^2$m$^{-3}$ and $\lambda_{3,anal} = 6.81 \times 10^{-26}$ kg$^3$m$^{-3}$.
**2.1 Initialisation**
In our test cases, all microphysical processes except collection are neglected and an exponential DSD
is initialised. In the results section, we will demonstrate that the outcome of the various collection
algorithms critically depends on how this initial, analytically defined, continuous DSD is translated
into a discrete ensemble of SIPs. Hence, the SIP initialisation is described in some detail.





### 151  2.1.1  SingleSIP-init and MultiSIP-init

First, the mass distribution is discretized on a logarithmic scale. The boundaries of bin $l$ are given
by $m_{bb,l} = m_{low} 10^{l/\kappa}$ and $m_{bb,l+1}$, where $m_{low}$ is the minimum droplet mass considered. The
bin centre is computed using the arithmetic mean $\bar{m}_{bb,l} = 0.5\,(m_{bb,l+1} + m_{bb,l})$. The bin size is
$\Delta m_{bb,l} = (m_{bb,l+1} - m_{bb,l})$. The mass increases tenfold every $\kappa$ bins. Several previous studies used
the parameter $s$ with $m_{bb,l+1}/m_{bb,l} = 2^{1/s}$ to characterise the bin resolution. The parameters $s$ and
$\kappa$ are related via $s = \kappa \log_{10}(2) \approx 0.3\,\kappa$.
For each bin, the droplet number is approximated by $\nu_b = f_m(\bar{m}_{bb,l})\,\Delta m_{bb,l}\Delta V$ and one SIP with
weighting factor $\nu_{sim} = \nu_b$ and droplet mass $\mu_{sim} = \bar{m}_{bb,l}$ is created, if $\nu_b$ is greater than a lower
cut-off threshold $\nu_{critmin}$. No SIP is created, if $\nu_b < \nu_{critmin}$. Moreover, no SIPs are created from
bins with radius $r < r_{critmin}$. We will refer to this as deterministic singleSIP-init. In its probabilistic
version, the mass $\mu_{sim}$ is randomly chosen within each bin $l$ and $\nu_{sim} = f_m(\mu_{sim})\,\Delta m_{bb,l}\Delta V$ is
adapted accordingly. By default, $r_{critmin} = 0.6\,\mu\text{m}$ and $\nu_{critmin} = \eta \times \nu_{max}$, which is determined
from the maximal weighting factor within the entire SIP ensemble $\nu_{max}$ and the prescribed ratio
of the minimal to the maximal weighting factor $\eta = 10^{-9}$. For larger $r_{critmin}$ it is advantageous to
initialise one additional "residual" SIP that contains the sum of all neglected contributions.
Following Unterstrasser and Sölch (2014, see their Appendix A), we introduce the multiSIP-init
technique. It is similar to singleSIP-init technique, except that we additionally introduce an upper
threshold $\nu_{critmax}$. If $\nu_b > \nu_{critmax}$ is fulfilled for a specific bin, then this bin is divided into $\kappa_{sub} =$
$\lceil \nu_b/\nu_{critmax} \rceil$ sub-bins and a SIP is created for each sub-bin. The multiSIP-init technique gives a
good trade-off between resolving low concentrations at the DSD tails and high concentrations of the
most abundant droplet masses.
So far, we introduced initialisation techniques with a strict lower threshold $\nu_{critmin}$ with no SIPs
created in bins with $\nu_b < \nu_{critmin}$. We can relax this condition by introducing—what we call—
a *weak* threshold. This means, that in such low contribution bin (with $\nu_b < \nu_{critmin}$) we create a
SIP with the probability $p_{create} = \nu_b/\nu_{critmin}$ and weighting factor $\nu_{sim} = \nu_{critmin}$. Having many
realisations of initial SIP ensembles, the expectation value of the droplet number represented by
such SIPs, $\nu_{critmin} \cdot p_{create} + 0 \cdot (1 - p_{create})$, equals the analytically prescribed value $\nu_b$. Using a
strict threshold the droplet number would be simply $0$ in those low contribution bins. In a related
problem, such a probabilistic approach has been shown to strongly leverage the sensitivity of ice
crystal nucleation on the numerical parameter $\nu_{critmin}$. This led to a substantial reduction of the
number of SIPs that are required for converging simulation results (Unterstrasser and Sölch, 2014).
Using the probabilistic version and a weak lower threshold is particularly important if different
realisations of SIP ensembles of the same analytic DSD should be created. The number of SIPs
$N_{SIP}$ depends on $\kappa$, $\nu_{critmin}, \nu_{critmax}$ and the parameters of the prescribed distribution.
Moreover, the singleSIP-init is used in a hybrid version, where different $\kappa$-values are used in
specified radius ranges.





### 2.1.2 $\nu_{const}$-init and $\nu_{draw}$-init

The accumulated PDF $F(m)$ is given by $\int_0^m \tilde{f}_m(m')dm'$ with the normalised PDF $\tilde{f}_m = f_m/\lambda_0$.
First, the size $N_{SIP}$ of the SIP ensemble that should approximate the initial DSD is specified. For
each SIP, its mass $\mu_i$ is reasonably picked by

$$\mu_i = F^{-1}(\text{rand}()),$$

where rand() generates uniformly distributed random numbers $\in [0, 1]$. In case of the $\nu_{const}$-init,
the weighting factors of all SIPs are equally $\nu_i = \nu_{const} = \mathcal{N}/N_{SIP}$. This init method reproduces
SIP ensembles similar to the ones in Shima et al. (2009) or Hoffmann et al. (2015). As a variety of
the $\nu_{const}$-init method, the weighting factors $\nu_i$ in the $\nu_{draw}$-init method are simply perturbed by

$$\nu_i = 2\,\text{rand}()\,\nu_{const}.$$

For the case of an exponential distribution, the following holds for the SIPs $i = 1, N_{SIP}$:

$$\mu_i = -\bar{m}\log(\text{rand}()).$$

In the literature, this approach is known as inverse transform sampling. A proof of correctness can
be found in classical textbooks, e.g. Devroye (1986, their section II.2).

### 2.1.3 $\nu_{random}$-init

The third approach allows specifying the spectrum of weighting factors that should be covered by
the SIP ensemble. Similar to the $\nu_{draw}$-init method, the weighting factors are randomly determined.
Whereas the latter method produced a SIP ensemble with weighting factors uniformly distributed
in $\nu$, the $\nu_{random}$-init produces weighting factors uniformly distributed in $\log(\nu)$ and covering the
range $[\mathcal{N}\,10^{\alpha_{low}}, \mathcal{N}\,10^{\alpha_{high}}]$. The eventual number of SIPs depends most sensitively on the param-
eter $\alpha_{high}$, which controls how big the portion of a single SIP can be.

SIPs with weighting factors $\nu_i = \mathcal{N}\,10^{(\alpha_{low}+(\alpha_{high}-\alpha_{low})\cdot\text{rand}())}$ are created, until $\sum_{j=1}^{N_{SIP}} \nu_j$ ex-
ceeds $\mathcal{N}$. The weighting factor of the last SIP is corrected such that $\sum_{j=1}^{N_{SIP}} \nu_j = \mathcal{N}$ holds. Now the
mass $\mu_i$ of each SIP is determined by the following technique: The first SIP represents the smallest
droplets and covers the mass interval $[0, m_1]$, whereas the last SIP represents the largest droplets in
the interval $[m_{N_{SIP}-1}, \infty]$. The SIPs $i$ in between cover the adjacent mass intervals $[m_{i-1}, m_i]$. The
boundaries are implicitly determined by $\int_0^{m_i} f_m(m')dm'\,\Delta V = \sum_{j=1}^i \nu_j$. The total mass contained
in each SIP is given by $\chi_i = \int_{m_{i-1}}^{m_i} f_m(m')m'dm'\,\Delta V$ and the single droplet mass by $\mu_i = \chi_i/\nu_i$.

For the case of an exponential distribution, the following holds for the interval boundaries and the
SIPs $i = 1, N_{SIP}$:

$$m_i = -\bar{m}\log\left(\mathcal{N} - \sum_{j=0}^i \nu_j\right)$$





and
$$\mu_i = \left( \frac{m_{i-1} - \bar{m}}{\exp(m_{i-1}/\bar{m})} - \frac{m_i - \bar{m}}{\exp(m_i/\bar{m})} \right) \frac{\mathcal{N}}{\nu_i}.$$
The above formulas must be carefully implemented such that numerical cancellation errors are kept
tolerable.
Experimenting with the SIP-init procedure, several optimisations have been incorporated. First,
the $\nu$-spectrum is split into two intervals $[\mathcal{N} 10^{\alpha_{low}}, \mathcal{N} 10^{\alpha_{med}}]$ and $[\mathcal{N} 10^{\alpha_{med}}, \mathcal{N} 10^{\alpha_{high}}]$. We
alternately pick random values from the two intervals. Without this correction, it happened that
several consecutive SIPs with small weights and hence nearly identical droplet masses are created,
which increases the SIP number without any benefits.
Going through the list of SIPs, the droplet masses increase and hence the individual SIPs contain
gradually increasing fractions of the total grid box mass. This can lead to a rather coarse repre-
sentation of the right tail of the DSD. Two options to improve this have been implemented. In the
$\nu_{random,rs}$-option, the $\nu_i$-values are reduced by some factor, that increases, as $\sum_{j=1}^{i} \nu_j$ approaches
$\mathcal{N}$. In the $\nu_{random,lb}$-option, $\nu$-values are randomly picked up to a certain radius threshold $r_{lb}$. Above
this threshold, SIPs are created with the singleSIP-method used on a linear bin.

### 2.1.4  Comparison

Figure 1 shows the weighting factors and other properties of the initial SIP ensemble, which may
affect the performance of the algorithms. Each column shows one class of initialisation techniques.
For a certain realisation, the first row shows the weighting factors $\nu_i$ of all SIPs as a function of their
represented droplet radius $r_i$. Each dot shows the $(\nu_i, r_i)$-pair of one SIP. For the singleSIP-init, the
dots are uniformly distributed along the horizontal axis, as one SIP is created from each bin (with
exponentially increasing bin sizes). The according $\nu$-values relate directly to the prescribed DSD.
The higher $f_m \Delta m$, the more droplets are represented in a SIP. No SIPs smaller than $r_{critmin} =$
$0.6 \, \mu$m are initialised and the $\nu$-values range over nine orders of magnitude consistent with $\eta =$
$10^{-9}$. The MultiSIP-init introduces an upper bound of $\nu_{critmax} = 2 \cdot 10^6$ for $\nu$. This threshold is
effective over a certain radius range where the SIPs have lower $\nu$-values compared to the singleSIP-
init and are also more densely distributed along the horizontal axis. For the $\nu_{const}$-init, all SIPs use
$\nu = \nu_{const}$, whereas for the $\nu_{draw}$-init the $\nu$-values scatter around this value. For $\nu_{const}$ and $\nu_{draw}$,
the $\nu$-values are chosen independently of the given DSD contrary to the latter techniques. However,
for both techniques, the density of the dots along the $r$-axis is correlated to $f_m \Delta m$.
The $\nu_{random}$-init technique randomly picks $\nu$-values which are distributed over a larger range
compared to the $\nu_{draw}$-init. In fact, they are uniformly distributed in $\log(\nu)$. The range of possi-
ble $\nu$-values can be adjusted and is chosen similar to the singleSIP/multiSIP by setting $\alpha_{high} =$
$-2, \alpha_{med} = -3$ and $\alpha_{low} = -7$. One possible advantage compared to the singleSIP-approach could
be that the occurrence of certain $\nu$-values is not limited to a certain radius range. In the singleSIP-





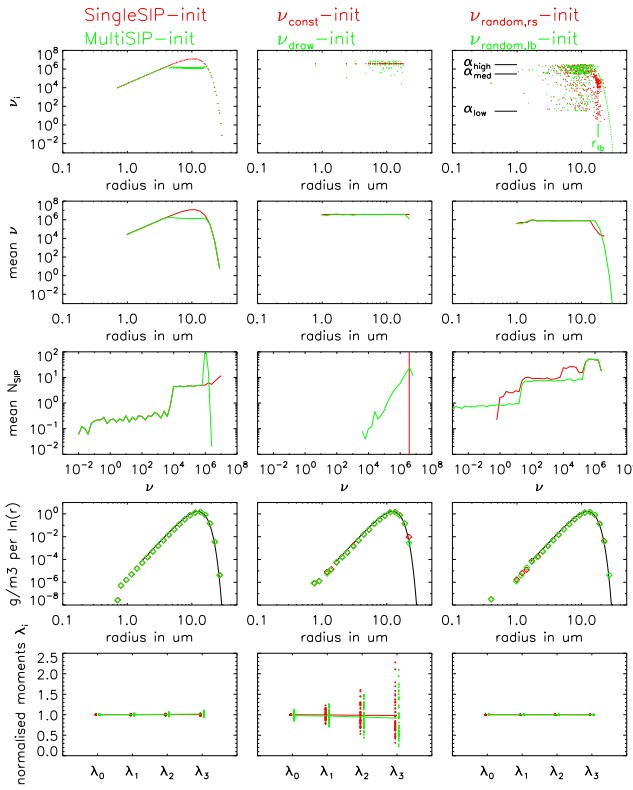

**Figure 1.** Characteristics of the various SIP initialisation methods (as given on top of each panel): Weighting factors $\nu_i(r_i)$ of an initial SIP ensemble, the mean weighting factors $\bar{\nu}(r)$, the occurrence frequency of the $\nu_i$-values and the resulting mass density distributions $g_{\ln r}$ are displayed (Row 1 to 4). Row 1 displays data of a single realisation, whereas rows 2 to 4 show averages over 50 SIP ensembles. The bottom row shows the moments $\lambda_0$, $\lambda_1$, $\lambda_2$ and $\lambda_2$ normalised by the respective analytical value. Every symbol depicts the value of a single realisation. The nearly horizontal line connects the mean values over all realisations. In the displayed examples, $\kappa = 20$ in the singleSIP-init, $\kappa = 20$, $\nu_{critmax} \approx 2 \cdot 10^6$ in the multiSIP-init, $N_{SIP} = 80$ in the $\nu_{const}$, $\nu_{draw}$-init and $(\alpha_{high}, \alpha_{med}, \alpha_{low}) = (10^{-2}, 10^{-3}, 10^{-7})$ in the $\nu_{random}$-inits (see black bars in top right panel). The vertical bar depicts the threshold radius $r_{lb}$.



init, the smallest $\nu$-values occur only at the left and right tail of the DSD, whereas in the $\nu_{random}$-
approach the smallest $\nu$-values (down to $\mathcal{N} 10^{\alpha_{low}}$) can appear over the whole radius range. The
horizontal bars in the plot indicate the values of $\alpha_{low}, \alpha_{med}$ and $\alpha_{high}$ and the vertical bar the
threshold radius $r_{lb}$.
The second row shows average $\nu$-value of all SIPs in a certain size bin. All init techniques are
probabilistic and the average is taken over 50 independent realisations of SIP ensembles. Not sur-
prisingly, the average $\nu$ of the $\nu_{draw}$-method is identical to $\nu_{const}$. Moreover, also for the $\nu_{random}$-
init the average $\nu$-value is constant over a large radius range. Only in the right tail, the $\nu$-values drop
as intended. The third row shows the occurrence frequency of weighting factors.
To display DSDs represented by a SIP ensemble, a SIP ensemble must be converted back into
a bin representation. For this, we establish a grid with resolution $\kappa_{plot} = 4$, count each SIP in its
respective bin, i.e. SIP $i$ with $m_{bb,l} < \mu_i \leq m_{bb,l+1}$ contributes to bin $l$ via $M_{bin,l} = M_{bin,l} + \mu_i \times \nu_i$
and $n_{bin,l} = n_{bin,l} + \nu_i$. We note that all displayed DSDs in this study will use $\kappa = 4$, irrespective of
the $\kappa$-value chosen in the initialisation. The fourth row shows such DSDs, again as an average over
50 SIP ensemble realisations. We find that any init technique is, in general, successful in producing
a meaningful SIP ensemble as the "back"-translated DSD matches the originally prescribed DSD
(black). Hence, the moments $\lambda_{k,SIP}$ match the analytical values $\lambda_{k,anal}$ for $0 \leq k \leq 3$, as shown in
the fifth row. Nevertheless for the $\nu_{const}$- and $\nu_{draw}$-init, the spread between individual realisations
can be large and they deviate substantially from the analytical reference. The singleSIP/multiSIP and
$\nu_{random}$, on the other hand, guarantee that each individual realisation is close to the reference.
**2.2 Description of Hypothetical algorithm**
First, we present a hypothetical algorithm for the treatment of collection/aggregation in an LCM,
which would probably yield excellent results. However is prohibitively expensive in terms of com-
puting power and memory, as $N_{SIP}$ increases drastically over time until the state is reached where
each SIP represents exactly one real droplet. Nevertheless, the presentation of this algorithm is useful
for introducing several concepts which will partly occur in the subsequently described "real-world"
algorithms.
Whereas condensation/deposition and sedimentation may be computed using interpolated quanti-
ties which implicitly assumes that all droplets of a SIPs are located at the same point, the numerical
treatment of collection usually assumes that the droplets of a SIP are spatially uniformly distributed,
i.e. well-mixed within the grid box. An approach, where the vertical SIP position is retained in the
collection algorithm and larger droplets overtaking smaller droplets is explicitly modelled, is de-
scribed in Sölch and Kärcher (2010), is not treated here.
Following Gillespie (1972) and Shima et al. (2009), the probability $P_{ij}$ that one droplet with mass
$m_i$ collides with one droplet with mass $m_j$ inside a small volume $\delta V$ within a short time interval $\delta t$





is given by:
$$P_{ij} = K_{ij}\,\delta t\,\delta V^{-1},\qquad(9)$$
where $K_{ij} = K(m_i, m_j)$.
For SIPs $i$ and $j$ containing $\nu_i$ and $\nu_j$ real droplets in a grid box with volume $\Delta V$, on average
$\nu_{coll} = P_{ij}\,\nu_i\,\nu_j$ collections between droplets from SIP $i$ and SIP $j$ occur. The average rate of such
$i-j$-collections ($i \neq j$) to occur is:
$$\frac{\partial \nu_{coll}(i,j)}{\partial t} = \nu_i\,K_{ij}\,\nu_j \Delta V^{-1} =: \nu_i o_{ij} =: O_{ij}.\qquad(10)$$
So-called self-collections, collisions of the droplets belonging to the same SIP ($i = j$), are described
by:
$$\frac{\partial \nu_{coll}(i,i)}{\partial t} = 2 \cdot \left(\frac{\nu_i}{2}\,K_{ii}\,\frac{\nu_i}{2}\Delta V^{-1}\right) = \frac{1}{2}\nu_i\,K_{ii}\,\nu_i \Delta V^{-1} =: \nu_i o_{ii} =: O_{ii},\qquad(11)$$
assuming that the SIP is split into two portions, each containing one half of the droplets of the original
SIP. The factor of 2 originates from the collections of each half, which have to be added to gain the
total number of self-collections for SIP $i$. Accordingly, the diagonal elements of the matrices $o_{ij}$ and
$O_{ij}$ differ from the off-diagonal elements by an additional factor of $0.5$. In terms of concentrations
(represented by SIPs in a grid box with volume $\Delta V$), we can write
$$\frac{\partial n_{coll}(i,j)}{\partial t} = K_{ij}\,n_i\,n_j\qquad(12)$$
for collections between different SIPs and
$$\frac{\partial n_{coll}(i,i)}{\partial t} = \frac{1}{2}\,K_{ii}\,n_i^2\qquad(13)$$
for self-collections.
In the hypothetical algorithm, the weighting factor of SIP $i$ is reduced due to collections with all
other SIPs and self-collections and reads as
$$\frac{\partial \nu_i}{\partial t} = -\sum_{j=1}^{N_{SIP}} \frac{\partial \nu_{coll}(i,j)}{\partial t} = -\sum_{j=1}^{N_{SIP}} O_{ij}.\qquad(14)$$
The droplet mass $\mu_i$ in SIP $i$ is unchanged.
For each $i-j$-combination, a new SIP $k$ is generated:
$$\frac{\partial \nu_k}{\partial t} = O_{ij} \quad\text{and}\quad \mu_k = \mu_i + \mu_j\qquad(15)$$
To avoid double counting only combinations with $i \geq j$ are considered.
The rate equations for the weighting factors can be numerically solved by a simple Euler forward
step: The weighting factor of existing SIPs is reduced by
$$\nu_i^{\Delta} := \left(\sum_{j=1}^{N_{SIP}} O_{ij}\right)\Delta t\qquad(16)$$





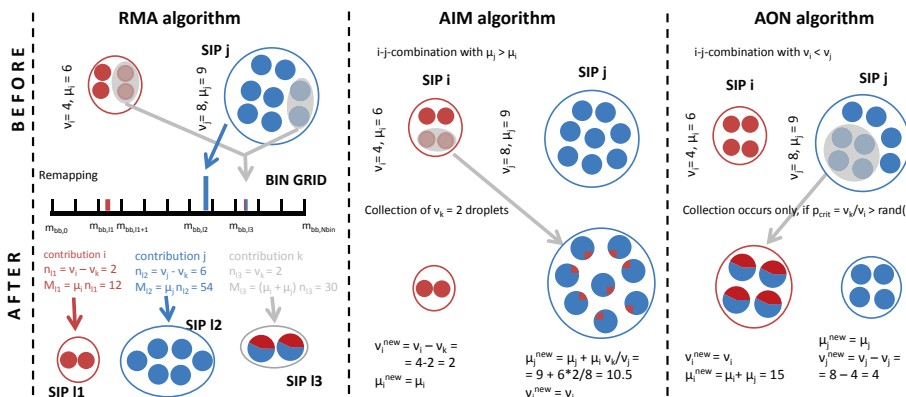

**Figure 2.** Treatment of a collection between two SIPs in the Remapping Algorithm (RMA), Average Impact Algorithm (AIM) and All-Or-Nothing Algorithm (AON).

leading to
$$\nu_i^* = \nu_i - \nu_i^\Delta,$$   (17)
or, equivalently,
$$\nu_i^* = \nu_i \left( 1 - \Delta t \sum_{j=1}^{N_{SIP}} o_{ij} \right).$$   (18)
For new SIPs $k$ we have
$$\nu_k = 0 + O_{ij} \cdot \Delta t.$$   (19)
Per construction the algorithm is mass-conserving subject to rounding errors.
In each time step, $N_{SIP,add} = N_{SIP}(N_{SIP} - 1)/2$ new SIPs are produced and the new number
of SIPs is $N_{SIP}^* = N_{SIP} + N_{SIP,add}$. After $nt$ time steps, the number of SIPs would be of order
$(N_{SIP,0})^{nt}$ which is not feasible.
In the following subsections, algorithms are presented that include various approaches to keep the
number of SIPs in an acceptable range.
In the following the various algorithms are described and pseudo-code of the implementations
is given. For the sake of readability, the pseudo-code examples show easy-to-understand imple-
mentations. The actual codes of the algorithms are, however, optimised in terms of computational
efficiency.
**2.3   Description of the Remapping (RMA) algorithm**



---

**Algorithm 1** Pseudo-code of the Remapping algorithm

---

1: *INIT BLOCK*

2: Given: Ensemble of SIPs;     Specify: $\kappa, \eta, \Delta t$

3: **for** $l = 1$ **to** $l_{max}$ **do** *{Create temporary bin}*

4:        $m_{bin,l} = m_{bin,low} 10^{l/\kappa}$

5: **end for**

6: *TIME ITERATION*

7: **while** t<Tsim **do**

8:        *LOSS BLOCK {Compute reduced bin contribution of existing SIPs}*

9:        **for** $i = 1$ **to** $N_{SIP}$ **do**

10:               Calculate $\nu_i^*$ according to Eq. 18

11:               Select bin $l$ with $m_{bb,l} < \mu_i \leq m_{bb,l+1}$

12:                   $n_{bin,l} = n_{bin,l} + \nu_i^*$

13:                   $M_{bin,l} = M_{bin,l} + \nu_i^* \cdot \mu_i$

14:        **end for**

15:        *GAIN BLOCK {Compute bin contribution of coalescing droplets}*

16:        **for all** $i < j \leq N_{SIP}$ **do**

17:               k++

18:               Compute $\nu_k$ according to Eq. 19

19:               $\mu_k = \mu_i + \mu_j$

20:               Select bin $l$ with $m_{bb,l} < \mu_k \leq m_{bb,l+1}$

21:                   $n_{bin,l} = n_{bin,l} + \nu_k$

22:                   $M_{bin,l} = M_{bin,l} + \nu_k \cdot \mu_k$

23:        **end for**

24:        *CREATE BLOCK {Replace SIPs}*

25:        Delete all SIPs

26:        **for all** $l$ with $M_{bin,l} > M_{critmin} = \eta\lambda_1$ **do** *{use $M_{critmin}$ as a weak threshold value}*

27:               i++

28:               Generate SIP $i$ with $\nu_i^{new} = n_{bin,l}$ and $\mu_i = M_{bin,l}/\nu_{bin,l}$

29:        **end for**

30:        $N_{SIP} = ii$

31:        $t = t + \Delta t$

32: **end while**

33: *EXTENSIONS*

34: *Self-collections for a kernel with $K(m,m) \neq 0$ can be easily incorporating in the algorithm by changing the condition in line 16 to $i \leq j \leq N_{SIP}$.*

---





First, the remapping algorithm is described as its concept follows closely the hypothetical algo-
rithm introduced in the latter section. The RMA algorithm is based on ideas of Andrejczuk et al.
(2010). We call their approach 'remapping algorithm' as $N_{SIP}$ is kept reasonably low by switch-
ing between a SIP representation and a bin representation in every time step. A temporary bin grid
with a pre-defined $\kappa$ is established which stores the total number $n_{bin,*}$ and total mass $M_{bin,*}$ of all
contributions belonging to a specific bin. The bin boundaries are given by $m_{bb,*}$.
Instead of creating a new SIP $k$ (with number $\nu_k$ obtained by Eq. 15 and mass $\mu_k = \mu_i + \mu_j$)
from each $i - j$-combination, the according contribution is stored on a temporary bin grid. More
explicitly, this means that the droplet number $n_{bin,l}$ of bin $l$ with $m_{bb,l} < \mu_k \leq m_{bb,l+1}$ is increased
by $\nu_k$. Similarly, the total mass $M_{bin,l}$ of that bin is increased by $\mu_k \nu_k$. Similarly, the reduced
contributions $\nu_i^*$ from the existing SIPs with droplet mass $\mu_i$ are added to their respective bins.
Figure 2 illustrates how a collection process between two SIPs is treated in RMA. In this example,
$\nu_k = 2$ droplets are produced by collection which have a droplet mass of $\mu_k = \mu_i + \mu_j = 15$. Instead
of creating a new SIP $k$ (as in the hypothetical algorithm), the contribution $k$ is recorded in the bin
grid. The droplet number $n$ in bin $l3$ is increased by $\nu_k = 2$ and the according total mass $M_{l3}$ by
$\nu_k \mu_k = 30$. The remaining contribution of SIP $i$ falls into bin $l1$ and $n_{l1}$ and $M_{l1}$ are increased by
$\nu_i^* = \nu_i - \nu_k = 2$ and $\mu_i \nu_i^* = 12$, respectively. The operation for SIP $j$ is analogous.
At the end of each time step after treating all possible $i - j$-combinations, a SIP ensemble is
created from the bin data with $\nu_i = n_{bin,l}$ and $\mu_i = M_{bin,l}/n_{bin,l}$.
Optionally, a lower threshold $\nu_{min,RMA}$ can be introduced, such that SIP $i$ is created only if
$n_{bin,l} > \nu_{min,RMA}$ holds. However, this may destroy the property of mass conservation which can
be remedied by the following.
We pick up the concept of a weak threshold introduced earlier and adjust it such that on av-
erage the total mass is conserved (instead of total number as before). We introduce the thresh-
old $M_{critmin} = \eta \lambda_1$. E.g. $\eta$ is set to $10^{-10}$, which implies that each SIP contains at least $10^{-10}$
of the total mass in a grid box. If $M_{bin,l} > M_{critmin}$, a SIP is created representing $\nu_i = n_{bin,l}$
drops with single mass $\mu_i = M_{bin,l}/n_{bin,l}$. If $M_{bin,l} < M_{critmin}$, a SIP is created with probability
$p_{create} = M_{bin,l}/M_{critmin}$. In this case the SIP represents $\nu_i = M_{critmin}/\mu_i$ droplets with single
mass $\mu_i = M_{bin,l}/n_{bin,l}$. Pseudo-code of the algorithm is given in algorithm 1.
Time steps typically used in previous collection/aggregation tests are around $\Delta t = 0.1$ to $10\,\text{s}$
depending inter alia on the used kernel. From Eq. 18 follows that the time step in RMA must satisfy
$$\Delta t < \sum_{j=1}^{N_{SIP}} o_{ij}. \tag{20}$$
Otherwise, negative $\nu$-values can occur which would inevitably lead to a crash of the simulation. In
mature clouds, the Long and Hall kernel attain large values which required tiny time steps of $10^{-4}\,\text{s}$
and smaller in the first test simulations. To be of any practical relevance, RMA had to be modified
in order to be able to run simulations with suitable time steps.





Hence, several extensions to RMA allowing larger time steps are discussed in the following.

1.   *Default version:*  Use the algorithm as outlined in Algorithm 1 (i.e. do not change anything).
     Negative $\nu_i^*$-values obtained by Eq. 17 are acceptable, as long as $n_{bin,l}$, from which the SIPs
     are created at the end of the time iteration, is non-negative for all $l$. This means that an existing
     SIP $i$ (which falls into bin $l$) can lose more droplets ($\nu_i^\Delta$) than it actually possesses ($\nu_i$) as long
     as the gain in bin $l$ (from all suitable SIP combinations) compensates this deficit. We will later
     see that this approach works well for the Golovin kernel, however fails for the Long and Hall
     kernel.

2.   *Clipping:*  Simply ignore bins with negative $n_{bin,l}$ and do not create SIPs from those bins.
     This approach destroys the property of mass conservation and is not pursued here.

3.   *Adaptive time stepping:* Instead of reducing the general time step, only the treatment of SIPs
     with $\nu_i^* < 0$ is sub-cycled. For each such SIP $i$, Eq. 17 is iterated $\tilde{\eta}_i$ times with time step
     $\Delta t_{SIP} = \Delta t / \tilde{\eta}_i$. Note that even though the computation of Eq. 17 and $O_{ij}$ involves the $\nu$-
     evaluation of all SIPs, only $\nu_i$ is updated in the subcycling steps and not the whole system of
     fully coupled equations is solved for a smaller time step. For sufficiently large $\tilde{\eta}_i$, $\nu_{i,subcycl}^*$ is
     positive, as $\nu_{i,subcycl}^\Delta < \nu_i$ as desired. Basically, we now assume that all collections involving
     SIP $i$ are equally reduced by a factor of $\eta_i = \nu_{i,subcycl}^\Delta / \nu_i^\Delta$ compared to the default time step.
     In the GAIN block of the algorithm (as termed in Alg. 1), all computations use the default
     time step and no sub-cycling is applied. To be consistent with the reduction in the LOSS
     block, Eq. 19 is replaced by $\nu_k = \eta_i O_{ij} \Delta t$.

4.   *Reduction limiter* The effect of an adaptively reduced time step can be reached with simpler
     and cheaper means. We introduce a threshold parameter $0 < \tilde{\gamma} < 1.0$ similar to the approach in
     Andrejczuk et al. (2012). Again, we focus on SIPs with $\nu_i^* < 0$ and simply set the new weight
     of SIP $i$ to $\nu_{i,RedLim}^* = \tilde{\gamma}\nu_i$. As above, all contributions involving SIP $i$ have to be re-scaled,
     now with $\gamma_i = (\nu_i - \nu_{i,RedLim}^*) / \nu_i^\Delta$.

5.   *Update on the fly* Another option to eliminate negative $\nu_i$-values is to do an "update on the
     fly". In this case, the algorithm is not separated in a LOSS and GAIN block. Instead, the $i-j$-
     combinations are processed one after another. After each collection process, as exemplified
     in Fig. 2, the weighting factors $\nu_i$ and $\nu_j$ of the two involved SIPs are reduced by $\nu_k$, i.e. the
     number of droplets that were collected. Subsequent evaluations of Eq. 19 then use updated $\nu$-
     values. Compared to the default version, it now matters in which order the $i-j$-combinations
     are processed, e.g. if you deal first with combinations of the smallest SIPs or of the largest
     SIPs.

## 2.4   Description of Average Impact (AIM) algorithm



---

**Algorithm 2** Pseudo-code of the average impact algorithm

---

1: ***INIT BLOCK + SIP SORTING***

2: Given: Ensemble of SIPs;     Specify: $\Delta t$

3: ***TIME ITERATION***

4: **while** t<Tsim **do**

5:       *{Sort SIPs by droplet mass}*

6:       Apply (adaptive) sorting algorithm, such that $\mu_j \geq \mu_i$ for $j > i$

7:       *{Compute total mass $\chi_i$ of each SIP}*

8:       $\chi_i = \nu_i \mu_i$

9:       **for** $i = 1$ **to** $N_{SIP}$ **do**

10:             *{Compute reduction of weighting factor due to number loss to all larger SIPs}*

11:             $\nu_i^{new} = \nu_i \left(1 - \Delta t \sum_{j=i+1}^{N_{SIP}} o_{ij}\right)$

12:             *{Compute mass transfer; mass gain from all smaller SIPs and mass loss to all larger SIPs}*

13:             $\chi_i^{new} = \chi_i - \chi_i \Delta t \sum_{j=i+1}^{N_{SIP}} o_{ij} + \sum_{j=1}^{i-1} \chi_j o_{ij} \Delta t$

14:       **end for**

15:       $\nu_i = \nu_i^{new}$

16:       $\mu_i = \chi_i^{new}/\nu_i^{new}$

17:       $t = t + \Delta t$

18: **end while**

19: ***EXTENSIONS***

20: *{Self-collections for a kernel with $K_{ii} \neq 0$ can be incorporated simply by adding the term $-0.5\,\Delta t\,o_{ii}$ inside the bracket on the r.h.s. of line 11 (see also Eq. (23) in the text)}*

---

The average impact algorithm by Riechelmann et al. (2012) and further developed in Maronga et al.
(2015) predicts the temporal change of the weighting factor, $\nu_i$, and the total mass of all droplets
represented by each SIP, $\chi_i = \nu_i \mu_i$. In this algorithm, two fundamental interactions of droplets are
considered (see also Fig. 7 in Maronga et al., 2015). First, the coalescence of two SIPs of different
size. It is assumed that the larger SIP collects a certain amount of the droplets represented by the
smaller SIP, which is then equally distributed among the droplets of the larger SIP. As a consequence,
the total mass and the weighting factor of the smaller SIP decrease, while the total mass of the larger
SIP increases accordingly. Fig. 2 illustrates how a collection between two SIPs is treated. SIP $j$ is
assumed to represent larger droplets than SIP $i$, i.e. $\mu_j > \mu_i$. As in the RMA example before, we
say that $\nu_k = 2$ droplets are collected. Then SIP $i$ loses two droplets to SIP $j$, i.e. $\nu_i$ is reduced by 2
and a mass of $\mu_i \nu_k$ is transferred to SIP $j$ where it is distributed among the existing $\nu_j = 8$ droplets.
Unlike to RMA, where droplets with mass $\mu_j + \mu_i = 15$ are produced, AIM predicts a droplet mass
of $\mu_j + \mu_i \nu_k/\nu_i = 10.5$ in SIP $j$. Usually, $\nu_k/\nu_i << 1$ and hence the name "average impact" for this
algorithm.



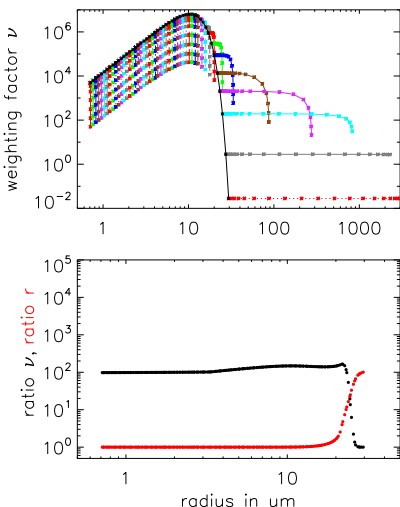

**Figure 3.** top: $(r_i, \nu_i)$-evolution of selected SIPs. The black line shows the initial distribution. Each coloured line connects the data points that depict the $(r_i, \nu_i)$-pair of an individual SIP every $200\,\mathrm{s}$.

bottom: Ratios $r_i(t = 3600\,\mathrm{s})/r_i(t = 0\,\mathrm{s})$ (red curve) and $\nu_i(t = 0\,\mathrm{s})/\nu_i(t = 3600\,\mathrm{s})$ (black curve) for all SIPs as a function their initial radius $r_i(t = 0\,\mathrm{s})$.

An example simulation with Long kernel, singleSIP-init, $\Delta t = 10\,\mathrm{s}, \kappa = 40$ and $N_{SIP} = 197$ is displayed.

Moreover, same-size collisions are considered in each SIP. This decreases the weighting factor of
each SIP but not its total mass. Accordingly, the radius of the SIP increases.
Both processes are represented in the following two equations which are solved for all colliding
SIPs (assuming that $\mu_0 \leq \mu_1 \leq \ldots \leq \mu_{N_{SIP}}$):
$$\frac{\mathrm{d}\nu_i}{\mathrm{d}t} = -K_{ii}\frac{1}{2}\frac{\nu_i\nu_i}{\Delta V} - \sum_{j=i+1}^{N_{SIP}} K_{ij}\nu_i\nu_j \Delta V^{-1} \qquad (21)$$

and
$$\frac{\mathrm{d}\chi_i}{\mathrm{d}t} = \sum_{j=1}^{i-1} \mu_j K_{ij}\nu_i\nu_j \Delta V^{-1} - \mu_i \sum_{j=i+1}^{N_{SIP}} K_{ij}\nu_i\nu_j \Delta V^{-1}. \qquad (22)$$

The first term on the right-hand-side of Eq. 21 describes the decrease of $\nu$ due to same-size col-
lections, the second term the decrease of $\nu$ due to collection by larger SIPs. The first term on the
right-hand-side of Eq. 22 describes the gain in total mass due to collections with smaller SIPs, while
the second term describes the loss of total mass due to collection by larger SIPs.
Using a Euler forward method for time integration the above equations read as:
$$\nu_i^{new} = \nu_i \left(1 - \sum_{j=i+1}^{N_{SIP}} o_{ij}\Delta t - 0.5\, o_{ii}\Delta t\right) \qquad (23)$$





and

$$\chi_i^{new} = \chi_i \left( 1 - \sum_{j=i+1}^{N_{SIP}} o_{ij} \Delta t \right) + \sum_{j=1}^{i-1} \chi_j o_{ij} \Delta t. \qquad (24)$$

Finally, the mass $\mu_i$ of each SIP is updated: $\mu_i^{new} = \chi_i^{new}/\nu_i^{new}$.
Figure 2 illustrates how the AIM algorithm works for an example simulation with the Long kernel
and singleSIP-init. The top panel shows the $(r_i, \nu_i)$-evolution of selected SIPs. The black line shows
the initial distribution. Each coloured line connects the data points that depict the $(r_i, \nu_i)$-pair of an
individual SIP every $200\,\mathrm{s}$. Clearly, $\nu_i$ of any SIP decreases over time, however the decrease is much
smaller for the largest SIPs and becomes zero for the largest SIP. The majority of SIPs starting from
the smallest radii show an opposite behaviour as their evolution is dominated by a strong $\nu_i$-decrease
at nearly constant $r_i$. In contrast, the evolution of the two largest SIPs is dominated by a strong $r_i$-
increase for constant $\nu_i$. The SIPs next to the largest SIPs undergo a transition; in the beginning, they
primarily grow in size, towards the end the decrease of $\nu_i$ is dominant. The bottom panel shows the
ratios $r_i(t = 3600\,\mathrm{s})/r_i(t = 0\,\mathrm{s})$ (red curve) and $\nu_i(t = 0\,\mathrm{s})/\nu_i(t = 3600\,\mathrm{s})$ (black curve) for all SIPs
of the simulation. Both ratios are smooth functions of the initial $r_i$ which is plotted on the $x$-axis.
By construction, the number of SIPs remains constant over the course of a simulation. Hence, the
number of SIPs per radius or mass interval decreases, when the DSD broadens over time. In our
example, the SIP resolution becomes coarser, particularly in the large droplet tail.
Negative values of $\nu_i^{new}$ and $\chi_i^{new}$ may occur. However, this case never occurred in our manifold
tests of the algorithm. The behaviour appears more benign than in RMA. Moreover, we found that
the algorithm preserved the initial size-sortedness of the SIP ensemble. However, for an arbitrary
kernel function and initial SIP ensemble, this is not guaranteed and we recommend to use adaptive
sorting algorithms that benefit from partially pre-sorted data sets (Estivill-Castro and Wood, 1992).
Adaptive sorting is also advantageous, when AIM is employed in real world applications, where
sedimentation, advection and condensation changes the SIP ensemble in each individual grid box.
**2.5   Description of the All-Or-Nothing (AON) algorithm**
The All-Or-Nothing (AON) algorithm is based on the ideas of Sölch and Kärcher (2010) and
Shima et al. (2009). Fig. 2 illustrates how a collection between two SIPs is treated. SIP $i$ is assumed
to represent fewer droplets than SIP $j$, i.e. $\nu_i < \nu_j$. Each real droplet in SIP $i$ collects one real droplet
from SIP $j$. Hence, SIP $i$ contains $\nu_i = 4$ droplets, now with mass $\mu_i + \mu_j = 15$. SIP $j$ now contains
$\nu_j - \nu_i = 8 - 4 = 4$ droplets with mass $\mu_j = 9$. Following Eq. 19, only $\nu_k = 2$ pairs of droplets would,
however, merge in reality. The idea behind this probabilistic AON algorithm is that such a collection
event is realised only under certain circumstances in the model, namely such that the expectation
values of collection events in the model and in the real world are the same. This is achieved if a
collection event occurs with probability $p_{crit} = \nu_k/\nu_i$ in the model. Then, the average number of





---

**Algorithm 3** Pseudo-code of the all-or-nothing algorithm; rand() generates uniformly distributed random numbers $\in [0,1]$.

---

1:   *INIT BLOCK*

2:   Given: Ensemble of SIPs;     Specify: $\Delta t$

3:   *TIME ITERATION*

4:   **while** t<Tsim **do**

5:       *{Check each $i-j$-combination for a possible collection event}*

6:       **for all** $i < j \leq N_{SIP}$ **do**

7:           Compute $\nu_k$ according to Eq. 15

8:           $\nu_{new} = \min(\nu_i, \nu_j)$

9:           $p_{crit} = \nu_k / \nu_{new}$

10:          *{Update SIP properties on the fly}*

11:          **if** $p_{crit} > 1$ **then**

12:               *MULTIPLE COLLECTION*

13:               *{can occur when $\nu_i$ and $\nu_j$ differ strongly and be regarded as special case; see text for further explanation}*

14:               assume $\nu_i < \nu_j$, otherwise swap $i$ and $j$ in the following lines

15:               *{$p_{crit} > 1$ is equivalent to $\nu_k > \nu_i$}*

16:               *{transfer $\nu_k$ droplets with $\mu_j$ from SIP $j$ to SIP $i$, allow multiple collections in SIP $i$, i.e. one droplet of SIP $i$ collects more than one droplet of SIP $j$.}*

17:               SIP $i$ collects $\nu_k$ droplets from SIP $j$ and distributes them on $\nu_i$ droplets: $\mu_i = (\nu_i\,\mu_i + \nu_k\,\mu_j)/\nu_i$

18:               SIP $j$ loses $\nu_k$ droplets to SIP $i$: $\nu_j = \nu_j - \nu_k$

19:          **else if** $p_{crit} >$rand() **then**

20:               *RANDOM SINGLE COLLECTION*

21:               assume $\nu_i < \nu_j$, otherwise swap $i$ and $j$ in the following lines

22:               *{transfer $\nu_i$ droplets with $\mu_j$ from SIP $j$ to SIP $i$}*

23:               SIP $i$ collects $\nu_i$ droplets from SIP $j$: $\mu_i = \mu_i + \mu_j$

24:               SIP $j$ loses $\nu_i$ droplets to SIP $i$: $\nu_j = \nu_j - \nu_i$

25:          **end if**

26:       **end for**

27:       $t = t + \Delta t$

28:   **end while**

29:   *EXTENSIONS*

30:   *{Self-collections for a kernel with $K(m,m) \neq 0$ can be treated in the following way: }*

31:   *{Insert the following loop before line 6 or after line 26.}*

32:   **for** $i = 1$ **to** $N_{SIP}$ **do**

33:       $p_{crit} = \nu_k / \nu_i$

34:       **if** $2\,p_{crit} >$rand() **then**

35:          *{every two (identical) droplets coalesce}*

36:          $\nu_i = \nu_i / 2$

37:          $\mu_i = 2\,\mu_i$

38:       **end if**

39:   **end for**

---




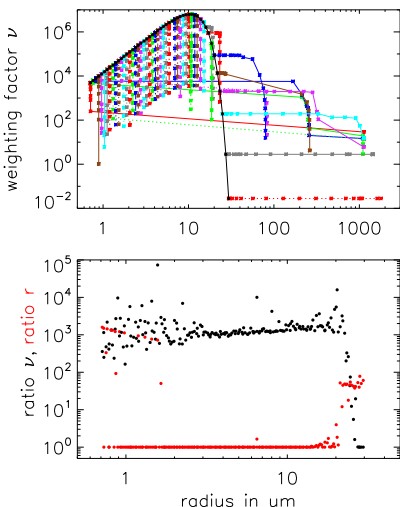

**Figure 4.** As in Fig. 3, for the AON algorithm.

collections in the model,
$\bar{\nu}_k = p_{crit}\nu_i = (\nu_k/\nu_i)\nu_i,$
is equal to $\nu_k$ as in the real world. A collection event between two SIPs occurs, if $p_{crit} >$rand(). The
function rand() provides uniformly distributed random numbers $\in [0,1]$.
Noticeably, no operation on the SIPs is performed if $p_{crit} <$rand().
The treatment of the special case $\nu_k/\nu_i > 1$ needs some clarification. This case is regularly en-
countered when the singleSIP-init is used, where SIPs with large droplets and small $\nu_i$ collect small
droplets from a SIP with large $\nu_j$. The large difference in droplet masses $\mu$ lead to large kernel
values and high $\nu_k$ with $\nu_i < \nu_k < \nu_j$. By the way, the case of $\nu_k$ being even larger than $\nu_j$ is not
considered, as it occurs only with unrealistically large time steps. If $p_{crit} > 1$, we allow multiple
collections, as each droplet in SIP $i$ is allowed to collect more than one droplet from SIP $j$. In total,
SIP $i$ collects $\nu_k$ droplets from SIP $j$ and distributes them on $\nu_i$ droplets. A total mass of $\nu_k\mu_j$ is
transferred from SIP $j$ to SIP $i$ and the droplet mass in SIPs $i$ becomes $\mu_i^{new} = (\nu_i\,\mu_i + \nu_k\,\mu_j)/\nu_i$.
The number of droplets in SIP $j$ is reduced by $\nu_k$ and $\nu_j^{new} = \nu_j - \nu_k$. Sticking to the example in
Fig. 2 and assuming $\nu_k = 5$, each of the $\nu_i = 4$ droplets would collect $\nu_k/\nu_i = 1.25$ droplets. The
properties of SIP $i$ and SIP $j$ are then: $\nu_i = 4$, $\mu_i = 17.25$, $\nu_j = 3$ and $\mu_j = 9$.
Another special case appears if both SIPs have the same weighting factor which regularly occurs
when the $\nu_{const}$-init is used. After a collection event, SIP $j$ would carry $\nu_j - \nu_i = 0$ droplets, whereas
SIP $i$ would still represent $\nu_i$ droplets. In this case, half of the droplets from SIP $i$ are moved to
SIP $j$ and both SIPs carry $\nu_j^{new} = \nu_i^{new} = 0.5\nu_i$ droplets with mass $\mu_i + \mu_j$. Without this correction,





zero-$\nu$ SIPs would accumulate over time and reduce the effective number of SIPs causing a poorer
sampling. Instead of this equal splitting, one can also assign unequal shares $\xi\nu_i$ and $(1-\xi)\nu_i$ to the
two SIPs (with $\xi$ being some random number).
Moreover, self-collections can be considered for kernels with $K_{ii} > 0$. If $2\,p_{crit} >$rand(), self-
collections occur between the droplets in a SIP (note the factor 2 due to symmetry reasons). Then
every two droplets within a SIP coalesce, implying $\nu_i = \nu_i/2$ and $\mu_i = 2\,\mu_i$.
So far, we explained how a single $i-j$-combination is treated in AON. In every time step, the full
algorithm simply checks each $i-j$-combination for a possible collection event. To avoid double-
counting only combinations with $i < j$ and self-collections with $i = j$ are considered. Pseudo-code
of the algorithm is given in Algorithm 3. The SIP properties are updated on the fly. If a certain SIP is
involved in a collection event in the model and changes its properties, all subsequent combinations
with this SIP take into account the updated SIP properties. Similar to the update on the fly version
of RMA, results may depend on the order in which the $i-j$-combinations are processed.
For most $i-j$-combinations, $p_{crit}$ is small and usually only a limited number of collection events
occurs in the model and AON may suffer from an insufficient sampling of the droplet space. Ac-
tual collections are a rare event in this algorithm. In our standard setup, $< 1\%$ of all possible col-
lections occur in the model until rain is initiated by very few lucky SIPs (similar to lucky drops,
e.g. Kostinski and Shaw (2005)). Indeed, Shima et al. (2009) reported convergence of AON only
for tremendously many SIPs (on the order of $10^5$ to $10^6$ in a box). We will later see that conver-
gence is possible with as few as $O(10^2)$ SIPs, if the SIPs are suitably initialised. Hence, it will
be demonstrated that AON is a viable option in 2D/3D cloud simulations, as already implied in
Arabas and Shima (2013).
As for AIM in Fig. 3, Fig. 4 (top) shows the $(r_i, \nu_i)$-evolution of selected SIPs for AON. The
picture looks more chaotic than for AIM, as each individual SIP has its own independent history due
to the probabilistic nature of AON. For the initially smallest SIP, only $\nu_i$ changes for most of the
time, as only collections occur where the partner SIPs have smaller weighting factors $\nu$. Towards
the end, the still very small SIP is at least once involved in a collection with a very large SIP that
has a larger $\nu$. Hence, $r_i$ of this SIP increases substantially. In contrast to the smallest SIP, other
initially small SIPs $i$ with similar properties are never part of a collection with $\nu_i < \nu_j$. Hence, their
radii $r_i$ remain small over the total period and $\nu_i$ is the only property that changes. The bottom panel
summarises the overall changes in $\nu_i$ (black) and $r_i$ (red) for all SIPs of the simulation. Unlike to
AIM, where only the initially largest SIPs grow, SIPs from both ends of the spectrum grow in AON.
Those SIPs have small $\nu$-values in common and in each collection their mass is updated to $m_i + m_j$.
The SIPs with initially large $\nu$-values lie in the radius range $[2\,\mu m, 15\,\mu m]$ and keep their initial radii
(at least in the singleSIP-init used here). The reductions in $\nu_i$ scatter around $\sim 10^3$ for most SIPs and
fall off to 1 for the largest SIPs.





For the generation of the random numbers, the well-proven (L'Ecuyer and Simard, 2007) Mersenne Twister algorithm by Matsumoto and Nishimura (1998) is used. AON simulations may be accelerated if random numbers are computed once a priori. However, this requires saving millions of random numbers for every realisation. An AON simulation with 1000 time steps and 200 SIPs implying $200 \times 100$ combinations, e.g. processes $2 \cdot 10^7$ random numbers. Using random numbers with a smaller cycle length deteriorated the simulation results in several tests and is not recommended.

The current implementation differs slightly from the version in Shima et al. (2009). Due to an unfavourable SIP initialisation similar to the $\nu_{const}$-technique, Shima et al. (2009) deal with large $N_{SIP}$-values in their simulations, where it becomes prohibitive to evaluate all $N_{SIP}(N_{SIP} - 1)$ SIP-combinations. Hence, they resort to $\lfloor N_{SIP}/2 \rfloor$ randomly picked $i-j$-combinations, where each SIP appears exactly in one pair (if $N_{SIP}$ is odd, one SIP is ignored). As only a subset of all possible combinations are numerically evaluated, the extent of collisions is underestimated. To compensate for this, the probability $p_{crit}$ is up-scaled with a scaling factor $N_{SIP}(N_{SIP} - 1)/(2 \lfloor N_{SIP}/2 \rfloor)$ to guarantee an expectation value as desired.

Moreover, in Shima's formulation the weighting factors are considered to be integer numbers. In contrast, we use real numbers $\nu$ which can even attain values below 1.0. This has several computational advantages: 1. better sampling of the DSD, in particular at the tails, 2. simpler AON implementation with fewer arithmetic and rounding operations, and 3. more flexibility, e.g. SIP splitting with real-valued $\xi$ in the case of identical weighting factors.

Sölch and Kärcher (2010) makes use of the vertical position of the SIPs and explicitly calculates whether or not a larger droplet overtakes a smaller droplet within a time step. This approach will be thoroughly analysed in a follow-up study.

In RMA and AIM SIPs with negative weights may be generated depending, e.g. on the condition $\Delta t \sum_{j=1}^{N_{SIP}} o_{ij} > 1$ in RMA. In AON, the latter condition implies that $\sum_{j=1} p_{crit,ij}$ of SIP $i$ is greater than unity. Hence, this SIP is likely to be involved in several collections (for $j$ with $p_{crit,ij} < 1$) or is involved in one or several multiple collections (for $j$ with $p_{crit,ij} > 1$).

## 3 Box model results

In this section, box model simulations of the three algorithms introduced in the latter section are presented, starting with the results of the Remapping (RMA) Algorithm, then those of the Average Impact (AIM) and finally the All-or-Nothing (AON) algorithm. The results of each algorithm are tested for three different collection kernels (Golovin, Long and Hall). Simulations with the Golovin kernel are compared against the analytical solution given by Golovin (1963). Consistent with many previous studies we choose $b = 1.5 \, \mathrm{m^3 \, kg^{-1} \, s^{-1}}$.

Simulations with the Long and Hall kernel are compared against high-resolution benchmark simulations obtained by the spectral-bin model approaches of Wang et al. (2007) and Bott (1998). In all





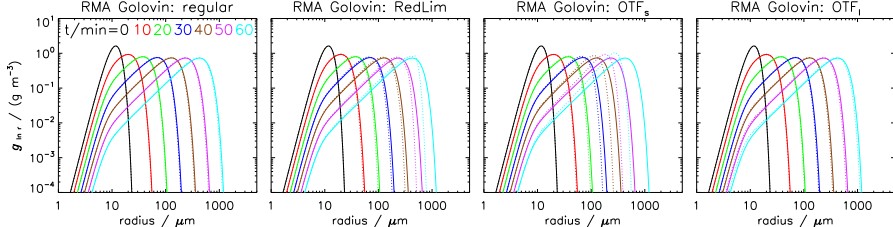

**Figure 5.** Mass density distributions obtained by the RMA algorithm for the Golovin kernel from $t = 0$ to $60\,\mathrm{min}$ every $10\,\mathrm{min}$ (from black to cyan). The solid curves show the reference solution, the dotted curves the simulation result of the RMA algorithm. The parameter settings are probabilistic singleSIP-init with weak threshold, $\kappa = 60$, $\eta = 10^{-8}$ and $\Delta t = 1\,\mathrm{s}$. The following versions of the RMA algorithm are depicted (from left to right): regular version, version with reduction limiter, version with update on the fly (start with combinations of smallest/largest droplets).

simulations, collision/coalescence is the only process considered in order to enable a rigorous evalu-
ation of the algorithms. The evaluation is based on the comparison of mass density distributions, and
the temporal development of 0th, 2nd, and 3rd moment of the droplet distributions. The 1st moment
is not shown since the mass is conserved in all algorithms per construction. As default, probabilistic
SIP initialisation methods are used. For each parameter setting, simulations are performed for 50
different realisations.

### 3.1 Performance of Remapping (RMA) Algorithm

Figure 5 compares DSDs of the RMA algorithm and the analytical reference solution for the Golovin
kernel. Each panel displays DSDs from $t = 0$ to $60\,\mathrm{min}$ every $10\,\mathrm{min}$. The left panel shows an ex-
cellent agreement of RMA with the reference solution and proves at least a correct implementation.
Figure 6 compares the temporal evolution of the moments. Moreover, the first row shows the number
of SIPs used in RMA. Except for the case with a very coarse grid ($\kappa = 5$) with fewer than 40 SIPs
in the end, the RMA results agree perfectly with the reference solution irrespective of the chosen $\kappa$
($\geq 10$) and minimum weak threshold $\eta$ ranging from $10^{-5}$ to $10^{-8}$. The number of non-zero bins
increases as the DSD broadens over time. In the last step of the time iteration, SIPs are created from
such bins. Hence, their number increases over time. Using a strict threshold, the total mass is not
conserved (not shown); The larger $\eta$ is, the more mass is lost. Hence, using a weak threshold or some
other measure (e.g. creation of a residual SIP containing contributions of all neglected bins) to avoid
this is highly recommended.
Next, RMA simulations with the Long kernel are discussed. As already mentioned, the default
RMA version would require tiny time steps which would rule out RMA from any practical appli-
cation. Both approaches introduced before, "Update on the fly" and "Reduction Limiter", succeed





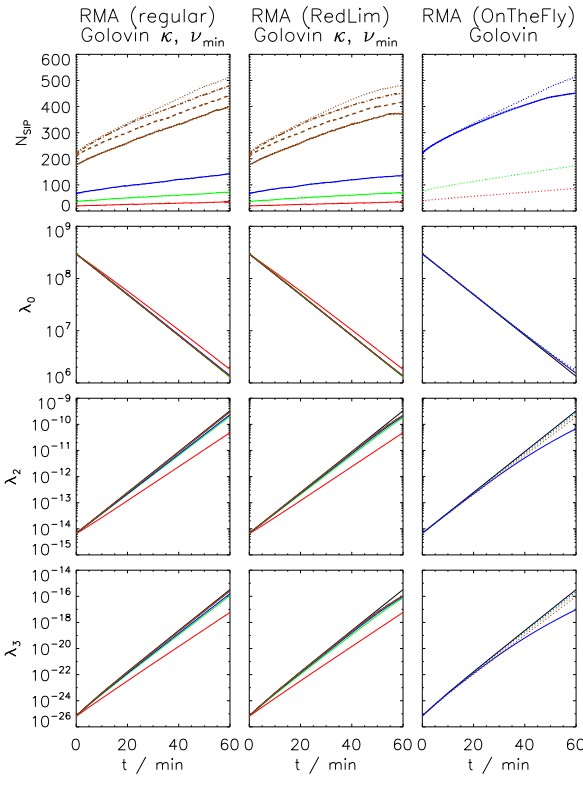

**Figure 6.** SIP number and moments $\lambda_0, \lambda_2$ and $\lambda_3$ as a function of time obtained by the RMA algorithm for the Golovin kernel. The black curves show the moments of the reference solution. All other curves depict the RMA results. The default settings are: Probabilistic singleSIP-init with weak threshold and $\Delta t = 1\,\mathrm{s}$. Left column: regular version with $\kappa = 60, 20, 10, 5$ (brown, blue, green, red) and threshold $\eta = 10^{-5}, 10^{-6}, 10^{-7}, 10^{-8}$ (solid, dashed, dash-dotted, dotted). Middle column: as in left column, but version with reduction limiter. Right column: version with update on the fly, solid/dotted lines: start with combinations of smallest/largest droplets, colours as before.





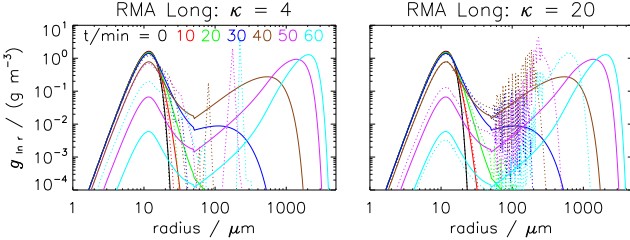

**Figure 7.** Mass density distributions obtained by the RMA algorithm for the Long kernel from $t = 0$ to $60\,\mathrm{min}$ every $10\,\mathrm{min}$ (from black to cyan). The solid curves show the reference solution, the dotted curves the simulation result of the RMA algorithm with Reduction Limiter ($\tilde{\gamma} = 0.1$). The parameter settings are probabilistic singleSIP-init with weak threshold, $\eta = 10^{-8}$, $\Delta t = 0.1\,\mathrm{s}$ and $\kappa = 4$ or $20$ (as indicated on top).

in eliminating negative $\nu_i$-values and in finishing the simulation within a reasonable time. However,
the results are not as desired. Fig. 7 shows the DSDs for a simulation with Reduction Limiter, weak
threshold $\eta = 10^{-8}$ and parameters $\kappa = 60$, $\Delta t = 0.1\,\mathrm{s}$ and $\tilde{\gamma} = 0.1$. Whereas the algorithm is ca-
pable of realistically reducing the number of the smaller droplets, it fails to predict the formation
of the rain mode and strong oscillations appear in the intermediate radius range $[100\,\mu\mathrm{m}, 200\,\mu\mathrm{m}]$.
We tested the algorithm with many parameter settings varying all of the aforementioned parameters,
$\Delta t \in [0.1\,\mathrm{s}, 1\,\mathrm{s}], \kappa \in [10, 60], \tilde{\gamma} \in [0, 1]$ and $\eta \in [10^{-10}, 10^{-5}]$ . Unfortunately, spurious oscillations
occur in most cases. Integrating over the whole mass spectrum, those oscillations do not average out
and, not surprisingly, the moments do not come close to the reference solution (not shown). Non-
oscillating results are obtained only if an unreasonably low resolution is used and very few bins exist
in the problematic radius range. However, in this case, the large droplet mode does not emerge and
the moments are again far from the reference. Hence, our RMA implementation is not capable of
producing reasonable results for the Long kernel.
It is not clear whether the oscillations are inherent to the original RMA algorithm or caused by the
introduction of the reduction limiter. The latter might introduce discontinuities where instabilities
could be triggered. The first option seems more probable, as the Golovin RMA simulations with
Reduction limiter do not show any instability and gives a perfect agreement with the reference (see
column 2 in Figs. 5 and 6). Similarly, Golovin RMA simulations with update on the fly are stable
and close to the reference, however the results depend on the order in which the SIP combinations
are processed (see column 3 (and 4) in Figs. 5 and 6). Again, Long simulations with an update on
the fly version of RMA are unstable (not shown).
Andrejczuk et al. (2010) introduced and evaluated the RMA algorithm and applied it in a simula-
tion of boundary layer stratocumulus. Our findings are seemingly in conflict with the conclusions of
their evaluation exercises. What both studies have in common is a similar trend for a $\kappa$-variation. In




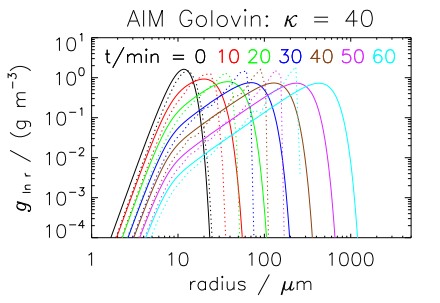
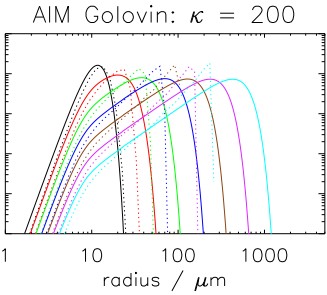

**Figure 8.** Mass density distributions obtained by the AIM algorithm for the Golovin kernel from $t = 0$ to $60\,\text{min}$ every $10\,\text{min}$ (from black to cyan). The solid curves show the reference solution, the dotted curves the simulation result of the AIM algorithm (ensemble average over 50 realisations). The parameter settings are: probabilistic singleSIP-init, $\nu_{critmin} = 10^{-9}\max(\nu_i)$, $\Delta t = 1\,\text{s}$ and $\kappa = 40$ (left) or $\kappa = 200$ (right).

their Fig. 13, simulations for $\kappa$ ranging roughly from $4$ to $30$ are depicted. The simulations with many
bins show oscillations, whereas the coarsest simulation has no oscillations, but is clearly far from
the real solution (largest droplets around $40\,\mu\text{m}$ compared to $500\,\mu\text{m}$ in the reference simulation).
In their Fig. 14, they presented a detailed sensitivity test only for a $\kappa = 4$ simulation, which down-
plays the severity of the oscillation issue. Moreover, their simulations ran up to $2000\,\text{s}$ compared to
$3600\,\text{s}$ in this study and many other studies (e.g. Bott, 1998; Wang et al., 2007). Hence, they missed
the regime where the effect of the oscillations is strongest. Despite our extensive tests we cannot
exclude that in Andrejczuk et al. (2010) an RMA implementation was used where oscillations are
less cumbersome; however, the study missed to demonstrate this for a conclusive test case and we
come to the conclusion that the evaluation exercises were incomplete and not suited to reveal the
deficiencies faced here.
RMA simulations with the Hall kernel are similarly corrupted by oscillations and do not produce
useful simulations either (not shown).

### 3.2   Performance of Average Impact (AIM) Algorithm

Fig. 8 displays DSDs obtained by AIM for the Golovin kernel. Compared to the reference, the
droplets pile up at too small radii and the algorithm is not capable of reproducing the continuous
shift to larger sizes, even if a fine grid with $\kappa = 200$ (right) instead of $\kappa = 40$ (left) is used. For both
$\kappa$-values, the increase of the higher moments proceeds at a too low rate (see Fig. 9), whereas the
decrease in droplet number matches the analytical evolution. AIM is a very robust algorithm in the
sense that the results are fairly insensitive to most numerical parameters as demonstrated for $\kappa$ and
$\Delta t$ in the left column of Fig. 9. Most simulations converge to—what we call—the best AIM solution,
which is, however, not the same as the correct solution. The results deteriorate slightly if the initial





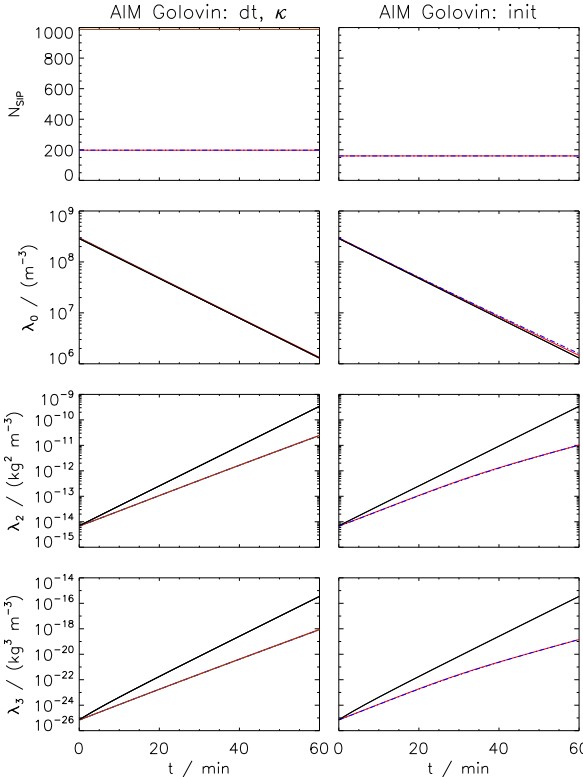

**Figure 9.** SIP number and moments $\lambda_0, \lambda_2$ and $\lambda_3$ as a function of time obtained by the AIM algorithm for the Golovin kernel. The black curves show the moments of the reference solution. All other curves depict the AIM results (average over 50 realisations). The default settings are: Probabilistic singleSIP-init, $\kappa = 40$, $\nu_{critmin} = 10^{-9} \max(\nu_i)$ and $\Delta t = 1\,\text{s}$. Left column: default simulation (red), larger time step ($\Delta t = 10s$, blue) and more SIPs ($\kappa = 200$, brown). Right column: $\nu_{const}$-init (red) and $\nu_{draw}$-init (blue) with $N_{SIP} = 160$.

SIP ensemble is generated with the $\nu_{const}$-init or $\nu_{draw}$-init instead of with the singleSIP-init (right
column of Fig. 9).
The algorithm performs, in general, better for the Long and Hall kernel as is detailed in the follow-
ing. Fig. 10 displays DSDs obtained by AIM for the Long kernel. Generally, the results are in good
agreement with the reference solution, as long as the SIP ensemble is initialised with the singleSIP-
init method (left and middle column). Towards the end of the simulated period (magenta and cyan
lines), the removal of small droplets is a bit underestimated and too many small droplets are present.
For $t = 30$ and $40\,\text{min}$, the large droplet mode is too weak as not enough large droplets have formed.
At that stage, the droplets grow rapidly by collection and the AIM results lag behind. Although the
offset is less than five minutes, it might become crucial in simulations of short-lived clouds. Also





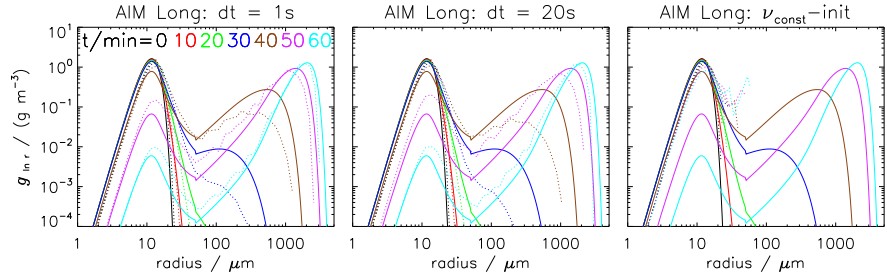

**Figure 10.** Mass density distributions obtained by the AIM algorithm for the Long kernel from $t = 0$ to $60\,\mathrm{min}$ every $10\,\mathrm{min}$ (from black to cyan). The solid curves show the reference solution, the dotted curves the simulation result of the AIM algorithm as an average over 50 realisations. The default settings are: Probabilistic singleSIP-init, $\kappa = 40$, $\nu_{critmin} = 10^{-9} \max(\nu_i)$, $\Delta t = 1\,\mathrm{s}$ (column 1); $\Delta t$ increased to $20\,\mathrm{s}$ (column 2); $\nu_{const}$-init technique with $N_{SIP} = 160$ (column 3).

the evolution of the moments (see Fig. 11) confirms this, as the onset of the rapid changes at around
$t = 30\,\mathrm{min}$ is only slightly retarded if parameters are suitably chosen. Towards the end, the AIM re-
sults get again very close to the reference solution. The left column of Fig. 11 shows the dependence
on the time step. For time steps $\Delta t \leq 20\,\mathrm{s}$ all results are similar to the best AIM solution which is
close to the reference. Time steps of $50\,\mathrm{s}$ and more do not produce good enough results. Moreover,
AIM is fairly insensitive to the choice of $\kappa$, $r_{critmin}$ and $\nu_{critmin}$ (see middle column). Simulations
with $\kappa$ ranging from 10 to 100 yield similar results. Only, for a very coarse resolution ($\kappa = 5$) with
25 SIPs, the decrease in droplet number is too small. Increasing the lower cutoff radius $r_{critmin}$
from $0.6\,\mu\mathrm{m}$ to $5\,\mu\mathrm{m}$, the $r < 5\,\mu\mathrm{m}$-part of the DSD is represented by a single SIP and $N_{SIP}$ is re-
duced by $60\%$. The predicted moments are unaffected by this variation. Those small-$r_i$ SIPs are not
relevant for the AIM performance. They simply carry too small fractions of the total grid box mass
to be important. Their status will not change over time as already illustrated in Fig. 3. Similarly, a
variation of $\nu_{critmin}$ or the switch to a strict threshold $\nu_{critmin}$ has no effect.
Now we draw the attention to the importance of the SIP-init method. The right panel of Fig. 10
shows the DSDs when the SIPs are initialised with the $\nu_{const}$-init method. The algorithm completely
fails and no droplets larger than $70\,\mu\mathrm{m}$ occur after 60 minutes. Consequently, the moments are far off
from the reference solution (solid lines in the right column of Fig. 11). Switching to the $\nu_{draw}$-init
method (dotted lines) or using many more SIPs (up to 1600) improves the results, yet they are still
useless. This clearly demonstrates how crucial the initial characteristics of the SIP ensemble are.
Initialising the SIPs with an appropriate technique like the singleSIP-init, useful results are obtained
with as few as 50 SIPs. Using the $\nu_{const}$-init or $\nu_{draw}$-init, on the other hand, solutions are still
useless, even though the number of SIPs and the computation time are factor 30 and 900 higher.





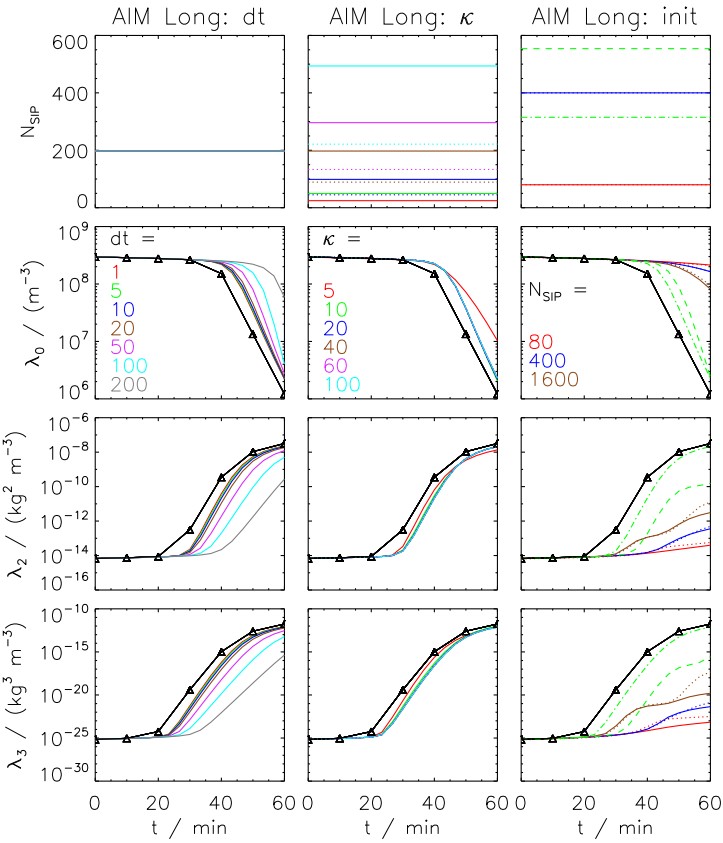

**Figure 11.** SIP number and moments $\lambda_0, \lambda_2$ and $\lambda_3$ as a function of time obtained by the AIM algorithm for the Long kernel. The black curves show the moments of the reference solution. All other curves depict the AIM results (average over 50 realisations). The left column shows a variation of $\Delta t = 1, 5, 10, 20, 50, 100, 200\,\text{s}$ for $\kappa = 40$. The middle column a variation of $\kappa = 5, 10, 20, 40, 60, 100$ for $\Delta t = 10\,\text{s}$. Either, the default singleSIP-init (solid) or the singleSIP-init with $r_{critmin} = 5\,\mu\text{m}$ (dotted) is used. The right column displays simulations with different initialisation techniques and $\Delta t = 10\,\text{s}$: the $\nu_{const}$-init (solid) and $\nu_{draw}$-init (dotted) with $N_{SIP} = 1600, 400, 80$ as well as the $\nu_{random,rs}$-init (dashed) and $\nu_{random,lb}$-init (dash-dotted) with $(\alpha_{high}, \alpha_{med}, \alpha_{low}) = (10^{-2}, 10^{-3}, 10^{-13})$ and threshold radius $r_{lb} = 16\,\mu\text{m}$.




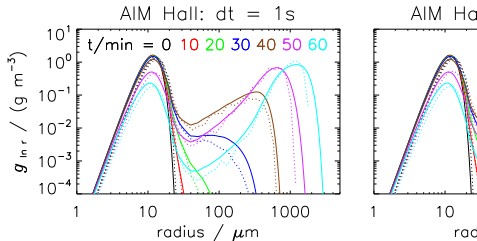

**Figure 12.** Mass density distributions obtained by the AIM algorithm for the Hall kernel from $t = 0$ to $60\,\mathrm{min}$ every $10\,\mathrm{min}$ (from black to cyan). The solid curves show the reference solution, the dotted curves the simulation result of the AIM algorithm as an average over 50 realisations. The default settings are: Probabilistic singleSIP-init, $\kappa = 40$, $\nu_{critmin} = 10^{-9}\max(\nu_i)$, $\Delta t = 1\,\mathrm{s}$ (column 1); $\Delta t$ increased to $20\,\mathrm{s}$ (column 2).

The $\nu_{random}$-simulations give another example of the importance of the init method. Even though
both techniques, $\nu_{random,rs}$ (dashed line) and $\nu_{random,lb}$ (dash-dotted line), are similar in design
and differ only in the creation of the largest SIPs (see Fig. 1), the outcome of the simulations is quite
different. For the $\nu_{random,lb}$-init, the solution matches the best AIM solution, whereas for $\nu_{random,rs}$
the moments $\lambda_2$ and $\lambda_3$ stagnate at too low levels. The latter test pinpoints the main weakness of the
AIM which is also reflected in its name (average impact). The initial weighting factors of those SIPs
(in relation to $\nu$ of the remaining SIPs) controls how strong this growth is and how the large droplet
mode emerges.
All quantities shown in Fig. 9 and 11 are averages over 50 realisations of the initial SIP ensem-
ble. All individual realisations yield basically identical simulation results and it would have been
sufficient to carry out and display simulations of a single realisation.
Figure 12 shows DSDs of simulations with the Hall kernel. Compared to the Long simulations,
small droplets are much more abundant (see reference solution), as the collection of small droplets
proceeds at a lower rate. This makes the simulation less challenging from a numerical point of view
and AIM DSDs come closer to the reference than in the Long simulations. Consequently, the AIM
moments agree very well with the reference as shown in Fig. 13. For $\Delta t \leq 20\,\mathrm{s}$ and $\kappa \geq 20$, all
solutions are similar to the best AIM solution.
**3.3   Performance of All-Or-Nothing (AON) Algorithm**
Fig. 14 shows the AON results for the Golovin kernel. An excellent agreement with the reference
solution is found which proves at least the correct implementation of AON. Switching to a version
without multiple collections (i.e. SIP $i$ collects at most $\nu_i$ droplets in every time step) does not affect
the solution as cases with $p_{crit} > 1 \Leftrightarrow \nu_k > \nu_i$ occur rarely. The AON moments closely follow the
reference solution, even when the time step is increased from $1\,\mathrm{s}$ to $10\,\mathrm{s}$ or fewer SIPs are used when
$\kappa$ is decreased from $40$ to $10$ (left column of Fig. 15). Unlike to AIM, AON is successful, even when




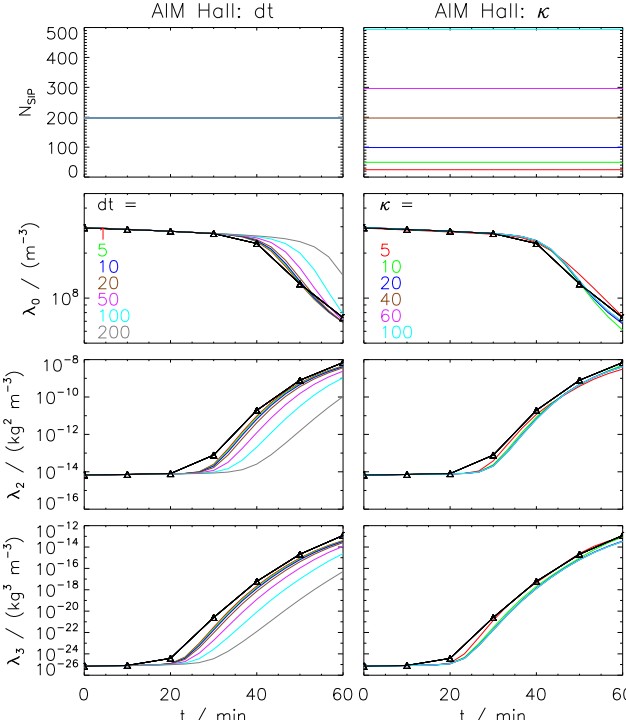

**Figure 13.** SIP number and moments $\lambda_0, \lambda_2$ and $\lambda_3$ as a function of time obtained by the AIM algorithm for the Hall kernel. The black curves show the moments of the reference solution. All other curves depict the AIM results (average over 50 realisations). The left column shows a variation of $\Delta t = 1, 5, 10, 20, 50, 100, 200\,\mathrm{s}$ for $\kappa = 40$ and the right column a variation of $\kappa = 5, 10, 20, 40, 60, 100$ for $\Delta t = 10\,\mathrm{s}$

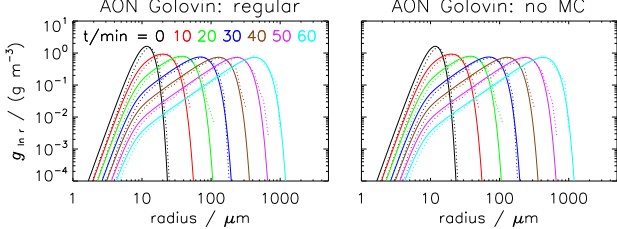

**Figure 14.** Mass density distributions obtained by the AON algorithm for the Golovin kernel from $t = 0$ to $60\,\mathrm{min}$ every $10\,\mathrm{min}$ (from black to cyan). The solid curves show the reference solution, the dotted curves the simulation result of the AON algorithm (ensemble average over 50 realisations). The parameter settings are: probabilistic singleSIP-init, $\kappa = 40$, $\nu_{critmin} = 10^{-9} \max(\nu_i)$, $\Delta t = 1\,\mathrm{s}$. The columns show various variants of the algorithm: default version, version disregarding multiple collections and version disregarding self-collections (from left to right).





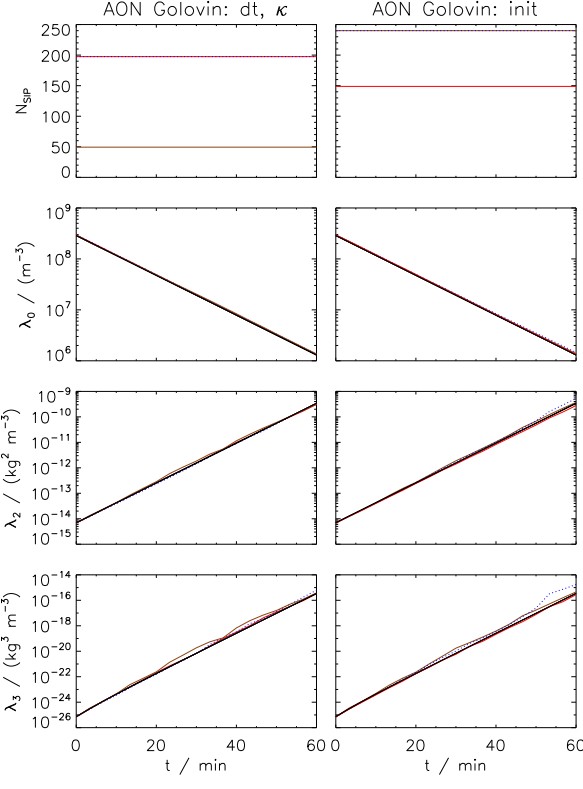

**Figure 15.** SIP number and moments $\lambda_0, \lambda_2$ and $\lambda_3$ as a function of time obtained by the AON algorithm for the Golovin kernel. The black curves show the moments of the reference solution. All other curves depict the AON results (average over 50 realisations). The default settings are: Probabilistic singleSIP-init, $\kappa = 40, \nu_{critmin} = 10^{-9}\max(\nu_i)$ and $\Delta t = 1\,\mathrm{s}$. Left column: default simulation (red), larger time step ($\Delta t = 20s$, blue) and fewer SIPs ($\kappa = 10$, brown). Right column: $\nu_{const}$-init (brown), $\nu_{draw}$-init (blue) and singleSIP-init with $r_{critmin} = 1.6\,\mu\mathrm{m}$ (red).

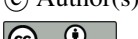



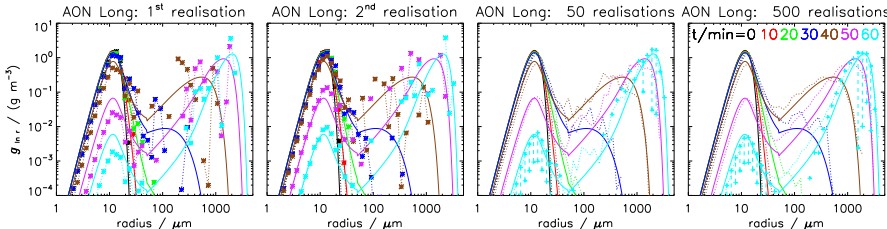

**Figure 16.** Mass density distributions obtained by the AON algorithm for the Long kernel from $t = 0$ to $60\,\mathrm{min}$ every $10\,\mathrm{min}$ (from black to cyan). The solid curves show the reference solution, the dotted curves the simulation result of the AON algorithm. Columns 1 and 2 show individual realisations (each $*$-symbol depict a non-zero $g$-value). Columns 3 and 4 show averages over 50 and 500 realisations. For each bin, the interquartile range is determined and depicted by $+$-symbols with a dashed bar (only for $t = 60\,\mathrm{min}$). If there is only one $+$-symbol, the 25th percentile is too small to be visible. The settings are: Probabilistic singleSIP-init, $\kappa = 40$, $\nu_{critmin} = 10^{-9}\max(\nu_i)$, $\Delta t = 20\,\mathrm{s}$.

the initial SIP ensemble is created with the $\nu_{const}$-init or $\nu_{draw}$-init (right column of Fig. 15). The
moments are averages over 50 realisations. For the $\nu_{draw}$-init method, the deviation in $\lambda_3$ towards
the end of the simulated period is due to a single outlier realisation where the initial values of the
moments $\lambda_2$ and $\lambda_3$ were already much higher than $\lambda_2$ and $\lambda_3$ of the reference solution. Column 2
of Fig. 1 already illustrated the large uncertainty of the initial values, which becomes increasingly
larger for higher order moments. Hence, this outlier behaviour is associated with a deficiency of the
init technique rather than being an algorithm-intrinsic feature.

688        Nevertheless, the simulations reveal large differences between individual realisations which de-

serves a closer inspection. Fig. 16 displays DSDs of AON for the Long kernel. The two left panels
show DSDs of single realisations. The $*$-symbol depicts the $g$-value for each bin. Those symbols
are connected by default. An interruption of the connecting line indicates one or more empty bins
($g = 0$) where no SIPs exist in this specific radius interval. This occurs frequently and the solutions
are full of spikes and irregularly over- and undershoot the reference solution, particularly in the large
droplet mode. The small droplet mode is underestimated in the first realisation and overestimated in
the second realisation. The advantages of AON become apparent when the DSDs are averaged over
many realisations as shown in columns 3 and 4. Then the DSDs come close to the reference solution
and the interquartile range indicates the broad envelope the individual realisations span around the
reference solution. Whereas the average over 50 realisations still has some fluctuations, the average
over 500 realisations produces a smooth solution. There are two sources that are potentially respon-
sible for the large ensemble spread: the probabilistic SIP initialisation and the probabilistic AON
approach. In a sensitivity test, 50 realisations are computed, all using the same SIP initialisation ob-
tained by a deterministic singleSIPinit. Figure 17 compares those simulations to regular simulations





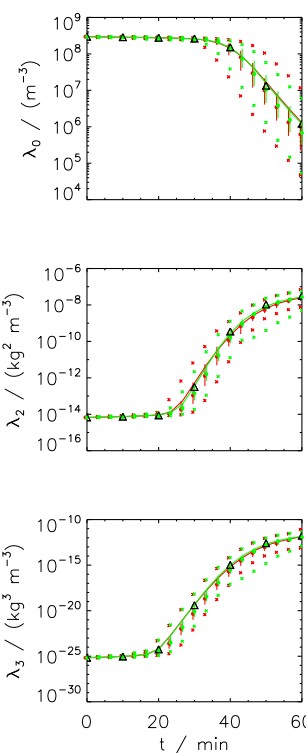

**Figure 17.** Moments $\lambda_0, \lambda_2$ and $\lambda_3$ as a function of time obtained by the AON algorithm for the Long kernel. Each realisation was initialised with a different SIP ensemble (probabilistic singleSIP, red) or all realisations started with the same SIP ensemble (deterministic singleSIP, green). In both cases, the curves show an average over 50 realisations with the vertical bars indicating the interquartile range. The crosses show the minimum and maximum values and the circle the median value. The black symbols depict the reference solution. The parameter settings are $\Delta t = 20$ and $\kappa = 40$.





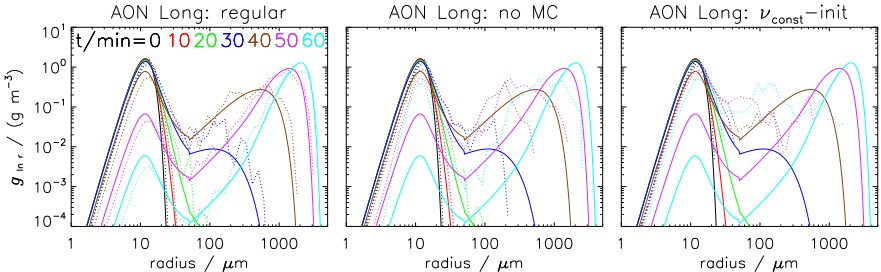

**Figure 18.** Mass density distributions obtained by the AON algorithm for the Long kernel from $t = 0$ to $60\,\mathrm{min}$ every $10\,\mathrm{min}$ (from black to cyan). The solid curves show the reference solution, the dotted curves the simulation result of the AON algorithm as an average over 50 realisations. The default settings are: Probabilistic singleSIP-init, $\kappa = 40$, $\nu_{critmin} = 10^{-9}\max(\nu_i)$, $\Delta t = 1\,\mathrm{s}$ (column 1); version disregarding multiple collections at $\Delta t = 10\,\mathrm{s}$ (column 2); $\nu_{const}$-init technique with $N_{SIP} = 160$ (column 3).

with differing SIP initialisations. In both cases, we find a substantial ensemble spread. Starting with
identical SIP initialisations the spread is, however, smaller suggesting that both sources contribute
to the ensemble spread.
Fig. 18 shows AON results with 50 realisations and probabilistic initialisation which gives a good
trade-off between computational cost and representativeness. Clearly, AON DSDs are less smooth
than those of AIM. Column 1 shows a default simulation with singleSIP init and shows very good
agreement with the reference solution. Disenabling multiple collections (column 2), far too few small
droplets become collected and their abundance is substantially overestimated. As a consequence, the
mass transfer from small to large droplets is slowed down and the large droplet mode is underesti-
mated. Using the $\nu_{const}$-init, the large droplet mode is not well matched and results are again useless.
Fig. 19 shows the temporal evolution of moments $\lambda_0, \lambda_2$ and $\lambda_3$ for a large variety of sensitivity tests.
Column 1 shows a variation of $\Delta t$ for the singleSIP-init. The larger $\Delta t$ is chosen, the more often
combinations with $p_{crit} > 1$ occur and the more crucial it becomes to consider multiple collections.
Even for the smallest time step considered, the version without multiple collections does not col-
lect enough small droplets and hence overestimates droplet number. With the regular AON version
considering multiple collections, reasonable results are obtained for time steps $\Delta t \leq 20\,\mathrm{s}$. Column 2
shows a variation of $\kappa$ for singleSIP-init. Whereas the higher moments perfectly match the reference,
the droplet number shows a non-negligible dependence on $\kappa$. For $\kappa < 100$, droplet number decrease
is faster, the finer the resolution is. For $\kappa > 100$, a variation of $\kappa$ has no effect, hence convergence is
reached. However, those simulations underestimate the droplet number. Best results are obtained for
an intermediate resolution of $\kappa = 40$. Using the MultiSIP-init, the simulations show the same unde-
sired behaviour. Hence, increasing the SIP concentration in the middle part of the initial DSD has no
positive effect despite using around $160\%$ more SIPs. In another experiment, the hybrid singleSIP-





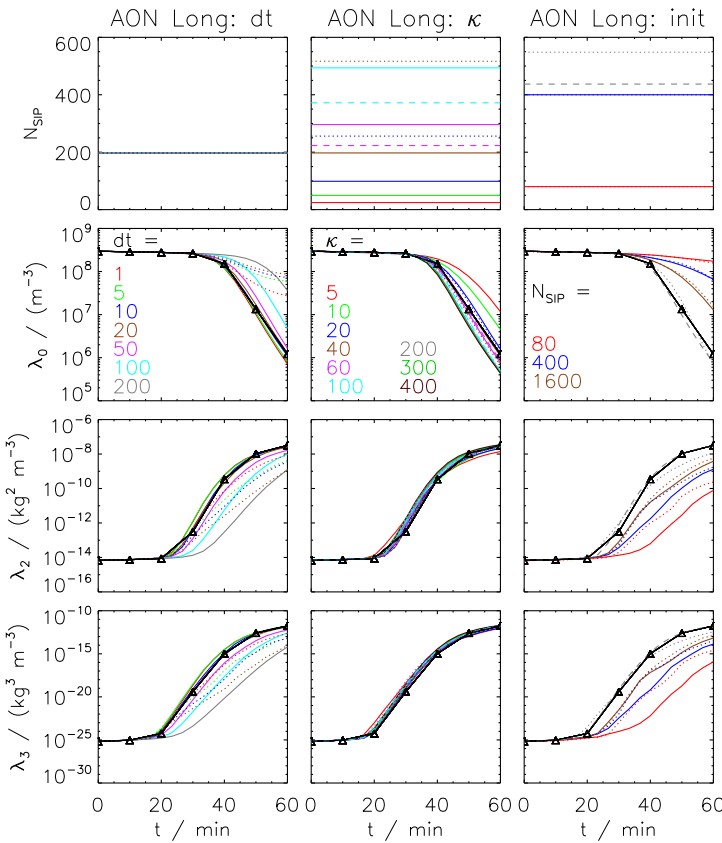

**Figure 19.** SIP number and moments $\lambda_0, \lambda_2$ and $\lambda_3$ as a function of time obtained by the AON algorithm for the Long kernel. The black symbols depict the moments of the reference solution. All coloured curves show the AON results (average over 50 realisations). The left column shows a variation of $\Delta t = 1, 5, 10, 20, 50, 100, 200\,\mathrm{s}$ for $\kappa = 40$ for the regular AON version (solid) and for a version disregarding multiple collections (dotted, only cases with $\Delta t \leq 20\,\mathrm{s}$ are displayed). The middle column shows a variation of $\kappa = 5, 10, 20, 40, 60, 100, 200, 300, 400$ for singleSIP-init (solid), singleSIP-init with $r_{critmin} = 1.6\,\mu\mathrm{m}$ (dashed, only for $\kappa = 60$ and $100$) and MultiSIP-init (dotted, only for $20 \leq \kappa \leq 100$). The right column shows simulations with the $\nu_{const}$-init (solid) and $\nu_{draw}$-init (dotted) with $N_{SIP} = 1600, 400, 80$. The gray dashed and dotted line show simulations with $\nu_{random,lb}$-init and $\nu_{random,rs}$-init, respectively. All simulations shown in the middle and right panel use $\Delta t = 10\,\mathrm{s}$.





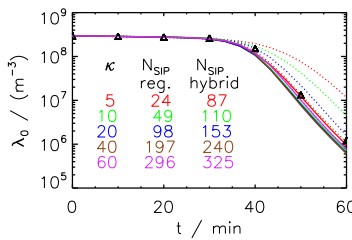

**Figure 20.** Droplet number as a function of time obtained by the AON algorithm for the Long kernel. The black symbols show the moments of the reference solution. The solid/dotted curves show simulations with hybrid/regular singleSIP-init for various $\kappa$-values (5 to 60, see legend). The hybrid version uses $\kappa = 100$ for radii above $15\,\mu$m and $\kappa$ as labeled for radii below $15\,\mu$m. The hybrid version uses more SIPs than the regular version (see $N_{SIP}$-values listed in the plot). The dotted lines are identical to solid lines in col 2 of 19

init was used. Below $r = 16\,\mu$m SIPs are initialised as usually. Above this radius, a high resolution
with $\kappa = 100$ is always used irrespective of the chosen $\kappa$. Clearly, more SIPs are initialised with this
hybrid version relative to the original version (see $N_{SIP}$-values listed in the figure legend). Figure 20
shows the droplet number evolution for the original singleSIP-init and the new hybrid version. The
sensitivity to $\kappa$ is basically suppressed when the hybrid version is used. This implies that the AON
algorithm is more or less insensitive to the resolution in radius range $r < 16\,\mu$m, however, it is sensi-
tive to the SIP resolution in the right tail. For example, the $\kappa = 5$-simulation with the hybrid version
and 87 SIPs performs better than the $\kappa = 20$-simulation with the regular init and 98 SIPs.
In the conventional version, SIPs are initialised down to a radius of $0.6\,\mu$m (as can be seen in
the top left panel of Fig. 1). Another variation of the singleSIP-init is shown in column 2 of Fig. 19
(dashed curves) where this lower cut-off radius is raised to $1.6\,\mu$m and around $25\,\%$ fewer SIPs
are used to describe the DSD. The simulation results are basically identical to the conventional init
version and suggest that those initially small-$r_i$, small-$\nu_i$ SIPs are not relevant for the performance
of AON.
Further tests with the singleSIP-init include a variation of the threshold parameter $\eta$ and a switch
from weak thresholds to strict thresholds. Moreover, we investigated the implications of update-on-
the-fly of the SIP properties. The singleSIP-init produces an initially radius-sorted SIP ensemble and
looping over the $i$-$j$ combinations in the algorithm starts with combinations of the smallest droplets,
which may introduce a bias. We reversed the order (i.e. started with largest droplet combinations) or
randomly rearranged the order of the SIP combinations. None of those variations had a significant
effect on the results (not shown).
Finally, the AON performance for other SIP initialisations is discussed (right column of Fig. 19).
As already demonstrated in Fig. 18, AON is not able to produce a realistic large droplet mode, if
a moderate number of SIPs is initialised with the $\nu_{const}$-technique. Hence, the higher moments are



underestimated and droplet number is overestimated. Increasing the number of SIPs up to $1600$,
the solutions get closer to the reference, yet the agreement is still not satisfactory. The performance
for the $\nu_{draw}$-init is similar. Keeping in mind the previous sensitivity studies (hybrid singleSIP-init,
MultiSIP-init), it is apparent that the $\nu_{const}$-init and $\nu_{draw}$-init suffer from an undersampling of
the initially largest droplets. Due to its simplicity, using constant weights for initialisation has been
a common approach in previous 3D-LCM cloud simulations (Shima et al., 2009; Hoffmann et al.,
2015). Hence, we tested AON extensions aiming at a better performance for equal weights ini-
tialisations. Let us consider the possible weighting factors the SIPs can attain in the course of a
simulation. In the beginning, all SIPs have $\nu = \nu_{init}$. After a collection event, for both involved SIPs
$\nu = \nu_{init}/2$. If such a $\nu = \nu_{init}/2$-SIP collects a $\nu = \nu_{init}$-SIP, both SIPs carry $\nu_{init}/2$ droplets.
Subsequent collections can generate SIPs with weighting factors $\nu_{init}/4$, $3\nu_{init}/4$ and so on. It may
be advantageous, if AON generates a broader spectrum of possible $\nu$-values and produces SIPs with
smaller weights more efficiently. So far, the equal splitting approach with $\xi = 0.5$ in a collection
event of two equal-$\nu$ SIPs has been used. In sensitivity tests, a random number for $\xi$ is drawn in
each collection event, either from a uniform distribution $\xi \in [0,1]$ or from a log-uniform distribution
$\xi \in [10^{-10}, 10^0]$. Enhancing the spread of $\nu$-values, more collection events occur in the algorithm,
as $p_{crit}$ is smaller when small-$\nu$ SIPs are involved. Once most SIPs were part of a collection event,
the first option with $\xi \in [0,1]$ produces a distribution of $\nu$-values that is similar to the initial $\nu$-
distribution of the $\nu_{draw}$-init technique. Hence, the new version does not improve the simulation
results, as the outcome for the $\nu_{draw}$-init and the standard $\nu_{const}$-init are similar (not shown). Other
variations produce smaller weights with $\xi = 10^{-10\,rand()}$ or $\xi = 10^{-10\,rand()^2}$, yet without any no-
ticeable improvement in the simulation results (not shown).
To complete the analysis for the Long kernel, the right column of Fig. 19 shows simulation results
for $\nu_{random,lb}$ and $\nu_{random,rs}$. In short, AON can cope with those initialisations and produces useful
results.
As already noted in the AIM section, Hall simulations are not as challenging as Long simulations
from a numerical point of view. As the collection of small droplets proceeds at a lower rate for the
Hall kernel, disenabling multiple collections in the AON simulations does not deteriorate the results
as much as in the Long simulations (not shown). Besides this, simulations with the Hall kernel lead
to similar conclusions as for the Long simulations and are therefore not discussed in more detail.
**4 Discussion**
The presented box model simulations can be regarded as a first evaluation step of collection/aggregation
algorithms in LCMs. The final goal is the evaluation in (multi-dimensional) applications of LCMs
with full microphysics. In order to isolate the effect of collection, other microphysical processes like
droplet formation and diffusional droplet growth have been switched off and all box model simula-





tions started with a prescribed SIP ensemble following a specific exponential distribution. The eval-
uation of different initialisation methods showed that the performance of the collection/aggregation
approaches depends essentially on the way the SIPs are initialised, a problem which is inherently
absent in spectral-bin models. Their initialisation resembles the singleSIP technique used here, i.e.
the number concentration (the weighting factor) within a bin (for a certain mass range represented
by one SIP) is directly prescribed. However, LCMs exhibit a larger variety of how an initial droplet
spectrum can be translated into the SIP space. The study showed that the singleSIP is advantageous
for the correct representation of the collisional growth, since they initialise large SIPs with small
weighting factors, which are responsible for the strongest radius growth. On the other hand, the
$\nu_{const}$ initialisation technique, in which all SIPs have the same weighting factor initially as it is
done in many current (multi-dimensional) applications of LCMs, impedes significantly the correct
representation of collisional growth.
In this idealised study, we were able to control (to a certain extent) the representation of droplet
spectra by various initialisation methods. In more-dimensional simulations with full microphysics,
however, this is not straightforward nor has it been intended. So far, convergence tests in "real-
world" LCM applications simply included variations of the SIP number and have not focused on
more detailed characteristics of the SIP ensemble (i.e. the properties that have been discussed in
Fig. 1). Droplet formation and diffusional droplet growth, which usually create the spectrum from
which collisions are triggered, should be implemented such that "good" SIP ensembles are gener-
ated or evolve before collection becomes important. Here, good refers to a SIP ensemble for which
the collection/aggregation algorithm performs well. For instance, the basic idea of the initialisation
technique $\nu_{random}$, the initialisation of weighting factors uniformly distributed in $\log(\nu)$, might also
improve multi-dimensional simulations.
Generally, the performance of the algorithms is better when the SIP ensemble features a broad
range of weighting factors. One viable option to achieve this is the introduction of a SIP splitting
technique (Unterstrasser and Sölch, 2014). Why this may improve the performance of the collec-
tion/aggregation algorithms is outlined next. Mass fractions represented by individual SIPs, $\tilde{\chi}_i$, are
analysed. $\tilde{\chi}_i$ is defined as $\chi_i/\mathcal{M}$, i.e. the total droplet mass in a SIP $\chi_i$ is normalised by the total
mass within the grid box $\mathcal{M}$. Figure 21 shows the initial $\tilde{\chi}_i$-values for the singleSIP-init method
and two resolutions $\kappa = 20$ and $100$ as a function of their initial radius $r_i$. The two rows show the
same data, using a logarithmic (top row) or linear $y$-scale (bottom). The log scale version highlights
that $\tilde{\chi}_i$-values spread over many orders of magnitudes. Mainly, the parameter $\nu_{critmin}$ controls the
minimum value of $\chi_i$. The heaviest SIPs carry initially up to $6.5\%$ ($\kappa = 20$) or $1.2\%$ ($\kappa = 100$) of
the total mass $\mathcal{M}$ (see bottom row). Clearly, the values of the $\kappa = 20$-simulation are larger, as the
total mass is distributed over fewer SIPs. For each SIP, $\tilde{\chi}_i$ is tracked over time and the maximum
value, $\tilde{\chi}_{i,max(t)}$, is recorded (red and brown curves in the graphs). Characteristically of AIM, only
the largest SIPs grow substantially and collect mass from other SIPs. Hence, only $\chi_i$ of those SIPs



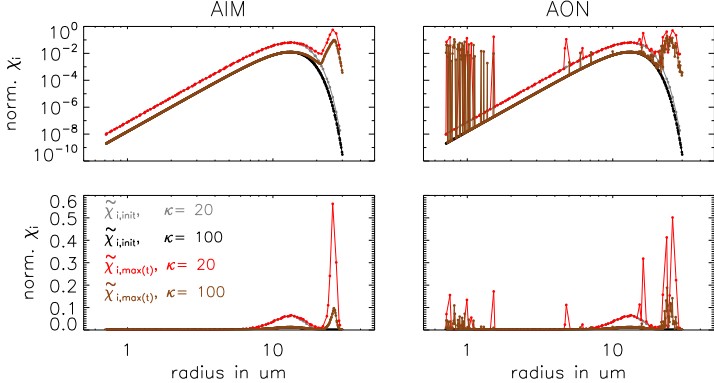

**Figure 21.** Normalised SIP mass $\tilde{\chi}_i$ as a function of the initial SIP radius $r_i$. $\tilde{\chi}_i$ is defined as $= \chi_i/\mathcal{M} = (\nu_i \mu_i)/\mathcal{M}$, i.e. the total droplet mass in a SIP is normalised by the total mass within the grid box. $\chi_{init}$ denotes $\tilde{\chi}_i$ of the initial SIP ensemble. $\chi_{max}$ denotes the maximum $\tilde{\chi}_i$-value each SIP attains over the course of a simulation. The left/right panel shows AIM/AON simulations with $\kappa = 20$ or $100$ (see legend). singleSIP-init, $\Delta t = 10\,\mathrm{s}$.

increases. By the way, this also illustrates that the $\chi_i$-values of the smallest SIPs are so small that
all those SIPs can be merged into a single SIP without changing the AIM outcome (see $r_{critmin}$-
variation before). Using the fine resolution ($\kappa = 100$), heavy SIPs carry up to $10\%$ of the total grid
box mass at some point in time. In the $\kappa = 20$-simulation, this ratio can be higher than $50\%$, meaning
that one specific SIP accumulated more than $50\%$ of the total grid box mass at some time. Hence, the
grid box mass is distributed fairly unevenly over the SIP ensemble. Astonishingly, this has no effect
on the performance of AIM as the predicted $\lambda_{k,SIP}$-values for both AIM simulations are basically
identical (see middle column of Fig. 11). In the AON simulations, we similarly find that the grid
box mass is unevenly distributed over the SIP ensemble. Different to AIM, also many initially small
SIPs and a few initially medium-sized SIPs carry a relevant portion of the grid box mass at some
time. The algorithms may converge better if those heavy SIPs are split into several SIPs during the
simulation.
In all simulations so far, the mean radius of the initial DSD was $9.3\,\mu\mathrm{m}$ where the abundance of
droplets larger than $10\,\mu\mathrm{m}$ drops strongly, which poses a challenge to the representation in SIP space.
In a sensitivity test, we start with "more mature" DSDs. The simulations are initialised with Wang's
reference solution after $t_{init} = 10$, $20$ or $30$ minutes (cf. red, green and blue solid curves in previous
plots of mass density distributions) using the singleSIP-init. Fig. 22 shows the SIP number and
various moments of the DSD for AIM and AON. The initial DSD is broader for a later initialisation
time and hence more SIPs are initialised for a given $\kappa$. This implies in particular that the spectrum
above $10-20\,\mu\mathrm{m}$ is sampled with more SIPs. For both algorithms, the simulation results are close to
the reference solution. Compared to the default $t_{init} = 0$-case, a much weaker $\kappa$-dependence of the





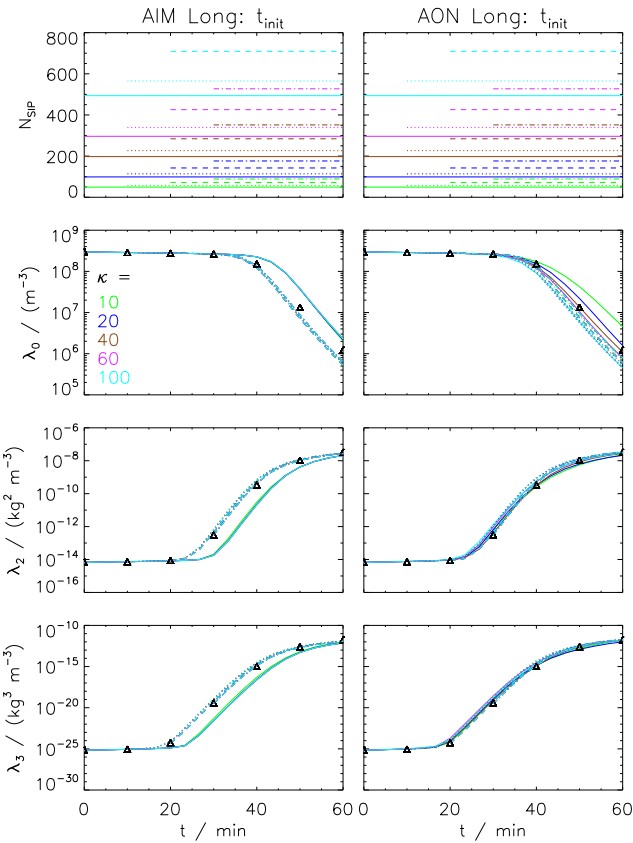

**Figure 22.** SIP number and moments $\lambda_0, \lambda_2$ and $\lambda_3$ as a function of time obtained for the Long kernel by AIM (left) and AON (right). The black symbols depict the moments of the reference solution. The simulations are initialised with Wang solution after 10 (dotted), 20 (dashed) or 30 (dash-dotted) minutes using the singleSIP-init with various $\kappa$-values (see legend). The default AON and AIM simulations initialised at $t = 0$, which have been shown before in Figs. 11 and 19, are depicted by solid lines.





AON predicted droplet number is apparent and the AIM results do not lag behind. Even though this
sensitivity test cannot be repeated for other init methods (as they require an analytical description of
the initial DSD), the singleSIP simulations already indicate that the SIP initialisation is not as crucial
when a later initialisation time is chosen and that our default setup with a narrow DSD may overrate
the importance of the SIP initialisation. What are the implications of this for simulations with full
microphysics? Clearly, the $t_{init} = 20\,\mathrm{min}$ and $30\,\mathrm{min}$-case oversimplify the problem, as such DSDs
cannot be produced by diffusional growth only. The $t_{init} = 10\,\mathrm{min}$-DSD, on the other hand, is still
close to the $t_{init} = 0\,\mathrm{min}$-DSD and may be produced by diffusional growth.
In multi-dimensional models, collection/aggregation might be further influenced by the movement
of SIPs due to sedimentation or flow dynamics. For instance, sedimentation removes the largest SIPs
with the smallest weighting factors, while turbulent mixing is able to add SIPs with their initial
weighting factor into matured grid boxes, where collection has already decreased the weighting
factors of the older SIPs. Indeed, the additional variability in more-dimensional simulations might
compensate for the missing variability in the weighting factors usually present in simulations using
the $\nu_{const}$ initialisation technique.
It is not clear which findings of our evaluation efforts are the most relevant aspects that control the
performance of collection/aggregation algorithms in more complex LCM simulations. Nevertheless,
the idealised box simulations are an essential prerequisite towards more comprehensive evaluations
as they disclosed the potential importance of the SIP initialisation (an aspect that is inherently absent
in spectral bin models). All in all, we can state that the behaviour of Lagrangian collection algorithms
in more complex simulations demands further investigation. Nevertheless, we have already learned
a lot from the box model simulations. A summary will be given in the concluding section.
Besides the academic Golovin kernel, our simulations used the hydrodynamic kernel with collec-
tion efficiencies that are usually employed for liquid clouds (Long and Hall). We found that Hall sim-
ulations are not as challenging as Long simulations from a numerical point of view. For ice clouds,
usually a constant aggregation efficiency $E_a$ (the analogon to collection efficiency) is chosen, partly
due to the lack of better estimates (Connolly et al., 2012). AON simulations with $E_a = 0.2$ indicated
that using a constant efficiency makes the computational problem less challenging, e.g. we find a
smaller sensitivity to $\kappa$ compared to the Long simulations shown in Fig. 19 (not shown). Hence, the
presented algorithms can be equally employed for aggregation. Certainly, the assumption of spheri-
cal particles used here is overly simplistic for ice cloud, in particular, if aggregates form. However,
including mass-area relationships (e.g. Mitchell, 1996; Schmitt and Heymsfield, 2010) in the ker-
nel expression and using parameterisations of ice crystal fall speed (e.g. Heymsfield and Westbrook,
2010) should not change the nature of the problem.



## 5 Conclusions

In the recent past, Lagrangian cloud models (LCMs), which use a large number of simulation particles (SIPs) to represent a cloud, have been developed and become more and more popular. Each SIP represents a certain number of real droplets, which is termed the weighting factor of a SIP. In particular, the collision process leading to coalescence of cloud droplets or aggregation of ice crystals is implemented differently in the various models described in the literature. The present study evaluates the performance of three different collection algorithms in a box model framework. All microphysical processes except collection/aggregation are neglected and an exponential droplet mass distribution is used for initialisation. The box model simulation results are compared to analytical solutions (in the case of the Golovin kernel) and to a reference solution obtained from a spectral bin model approach by Wang et al. (2007) (in the case of the Long or Hall kernel).

LCMs exhibit a large variety of how an initial droplet spectrum can be translated into the SIP space and various initialisation methods are thoroughly explained. The performance of the algorithms depends crucially on details of the SIP initialisation and various characteristics of the initialised SIP ensemble (an issue that is inherently absent in spectral bin models and has not been paid much attention in previous LCM studies).

The Remapping Algorithm (based on ideas of Andrejczuk et al., 2010) showed a poor performance, either no realistic rain mode developed or the solutions became unstable. The evaluation exercises presented in Andrejczuk et al. (2010) were not suited to reveal the obvious shortcomings or downplayed its severity. Based on our extensive tests, the algorithm cannot be recommended for further LCM applications, unless the stability issue is solved.

The Average Impact (AIM) algorithm (based on ideas of Riechelmann et al., 2012) can produce very good results, however, appears to be inflexible inasmuch as only the initially largest SIPs are allowed to grow in radius space. The performance depends on details of the SIP initialisation much more than, e.g. on the time step or the SIP number.

The probabilistic All-or-Nothing (AON) algorithm (based on ideas of Shima et al., 2009; Sölch and Kärcher, 2010) yields the best results and is the only algorithm that can cope with all tested kernels. Unlike to AIM, in AON it is not pre-determined which SIPs will eventually contribute to the large droplet mode. By design, any SIP can become significant at some point and the algorithm can cope with SIP initialisations that guarantee a broad spectrum of weighting factors. If an equal weights initialisation is used tremendously many SIPs are necessary for AON convergence as reported by (Shima et al., 2009). Many current (multi-dimensional) applications of LCMs use such SIP ensembles with a narrow spectrum of weighting factors causing a poor performance of the collection/aggregation algorithms. This should be clearly avoided in order to have collection/aggregation algorithms to work properly and/or efficiently. The time step and the bin resolution $\kappa$ (used in the singleSIP-init) have values similar to those used in traditional spectral-bin models and hence the computational efforts of both approaches for the collection/aggregation treatment are in the same range. The presented box



model simulations are a first step towards a rigourous evaluation of collection/aggregation algorithms
in more complex LCM applications (multidimensional domain, full microphysics).
**6   Code availability**
The programming language IDL was used to perform the simulations and produce the plots. The
source code can be obtained from the first author. Pseudo-code of the algorithms is given in the text.
**7   Competing interests**
The authors declare that they have no conflict of interest.
*Acknowledgements.* The DFG (German Science Foundation) partly funded the first author (contract number
UN286/1-2) and the second author (RA617/27-1). We thank A. Bott for providing us with his fortran code,
L.-P. Wang for simulation data, M. Andrejczuk, S. Shima and P. L'Ecuyer for discussions.





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
