# Peer review of "Collection/aggregation algorithms in Lagrangian cloud microphysical models: Rigorous evaluation in box model simulations"

_Geoscientific Model Development, 2016_

## Referee Comment (RC1) · Anonymous Referee #1 · 11 Jan 2017

The authors compare three different Langrangian Cloud Model (LCM) implementations with a focus on collection using three different collection kernels. Analytical solutions as well as previous bin model results are used as references. Additionally, sensitivity of the LCM implementations with respect to the initialization of the simulation particles (SIPs) as well as to different numerical features (resolution, time step, ...) is tested.

This results in a large amount of model runs with a great variety of possible parameter and configuration combinations which sometimes makes the manuscript difficult to read.

**General comments**

[Figure]

Each of the LCM implementations shows rather strong shortcomings:

- RMA cannot deal with realistic kernels (Long, Hall) and shows spurious oscillations.

- AIM systematically underestimates the collisional growth.

- AON always needs an ensemble of at least 50 realizations to reach a representative average result for the final drop size distribution since individual realizations deviate considerably from the average (in contrast to RMA and AIM). This severely limits the potential to be used in 2- and 3-dimensional models with a large number of grid points.

Additionally, sensitivities with respect to initialization of SIPs are shown to be high at least for some configurations. This problem is discussed towards the end of the manuscript where also more mature drop size distributions are used for initializations. Within a full microphysics description including drop nucleation and condensational growth, it should be harder to control the DSD at the moment when collisions become important. This discussion should be extended.

Compared to spectral bin models the accuracy of all LCM implementations shown seems to be lower with at least comparable computational costs. One could conclude that LCMs are of no practical use. Nevertheless, LCMs are valuable tools. Please discuss critically advantages and disadvantages of LCMs.

The quality of some figures is poor. Most of them are too small, lines are too thin and sometimes too many. Specific comments are given below.

**Specific comments**

l. 154-157: What is the reason to switch from mass doubling (which is often used) to a tenfold increase as a basis for bin resolution?

l. 238: If the probabilistic version of the singleSIP-init is used, dots are not distributed uniformly!

Fig. 1: upper left and below: difference between red and green lines is misleading since the higher density of dots wrt the x-axis is not resolved. upper right: threshold radius line barely can be seen; lines for alpha-values are also misleading, can be confused with legends. Values should read $N10^{\alpha}$. Last but one row: Is there a systematic difference between the symbols and the lines due to plotting issues? If not, a better initial agreement should be reached. Cp. also l. 268-270.

Algorithm 1: What do k++ and i++ stand for? loops over k/i?

Algorithm 2: l. 13: Please exchange gain and loss term due to consistency with l. 12 and eq. (22)

Fig. 3: Top: It is difficult to see what happens in the left part of the spectrum (<20 $\mu$m). Bottom: It is confusing to normalize one ratio to t=0 and the other one to t=3600. Please redo the black curve with $\nu_i(t=3600)/\nu_i(t=0)$

l. 559 and Figs. 6, 9, 11, 13, 15, 17, 19, 22: The third moment $\lambda_3$ should not be shown in the figures since the behaviour is very similar to $\lambda_2$ (which should be stated in the text). The space saved can be used to extend some other figures in order to improve their readability.

Fig. 5: Use full lines with enhanced line thickness for RMA results and dotted (or dashed) lines for analytical solutions. Otherwise, all plots look identical at first glance.

l. 564-575: The discussion of the RMA Golovin results is very short and misses several aspects, e.g.: Why are the results for RedLim worse than the regular ones? What are the reasons for the relatively large differences between the two OTF versions?

Fig. 6: Are there any lines missing? Variation of $\eta$ only for $\kappa = 60$ in the left and the middle column? No $\kappa = 60$ for OTF at all? Which lines fall together and which runs are carried out at all?

Fig. 7: see Fig. 5: full lines for the RMA results.

l. 595: Compared to the regular version (and to bin model results) I would not call the RMA RedLim results "perfect". The same holds for the $OTF_s$ results; only $OTF_l$ is almost "perfect".

Figs. 8, 10, and 12: see Fig. 5: full lines for the AIM results.

Fig. 14: see Fig. 5: full lines for the AON results. Results plot for disregarding self-collections is missing.

l. 739: Is this restricted to AON results or do the other methods show similar robustness wrt to the small tail of the distribution? In reality very small drops similarily do not contribute substantially to the growth of the large mode due to their low number and small individual mass. This should be reflected in model sensitivities.

l. 746: This is in contrast at least to the Golovin RMA results. Why is it reasonable for AON? Is this due to the lower number of collision events realised because of the probability restrictions?

l. 766: When $p_{crit}$ is smaller, less collection events can be expected (see lines 469/470). A spread in $\nu$-values leads to smaller and larger $\nu$-values. Does this mean, that the largest $\nu$-values are responsible for the enhanced collection?

l. 902ff: It should be critically mentioned that AON always needs an ensemble of at least 50 realizations to reach a representative average result for the final drop size distribution since individual realizations deviate considerably from the average (in contrast to RMA and AIM). This leads to a large effort in terms of computational resources.

**Technical corrections**

l. 18: ... are important processes ...

Table 1: mean mass: M/N

[Figure]

Fig. 1 caption: alpha should be (-2, -3, -7)

l. 256: values of $N10^\alpha$ ...

l. 276: However, it is ...

l. 371: rather "proposed" than "discussed"

l. 410ff.: The terms "larger SIP" and "smaller SIP" are used here with the meaning "SIP with larger/smaller drops(=higher average drop mass)". Please define whether "large SIP" indicates large drops or a large number of drops within the SIP (cp. l. 510).

Fig. 3 caption: ... function of their initial radius ... Please add that it is an AIM simulation.

l. 434: ... of each droplet within the SIP ...

l. 435: Figure 3

l. 510: In contrast to l. 410ff. smallest SIP refers to the size of the droplets not the weighting factors;

Fig. 11 caption: ... black curves with triangles ... green lines for $\nu_{random}$; $\alpha$ should be (-2, -3, -7)

l. 658: green lines

l. 792: check the meaning of "large SIP", also l. 824 "heavy SIP"

―――――――――――――

---

## Referee Comment (RC2) · A. Jaruga (Referee) · 13 Jan 2017

The Authors evaluate three algorithms for representing collisions in Lagrangian cloud microphysics schemes that are available in the literature. The design and some implementation details are discussed. The accuracy of the three collision algorithms is compared against the analytical solution of Golovin kernel and bin solutions of Long and Hall kernels. A very wide parameter space is investigated. The manuscript also tests three different initialization techniques for the Lagrangian cloud microphysics schemes.

The work presented here is very useful. Lagrangian schemes offer very detailed

representation of microphysical processes in clouds. Yet, because the Lagrangian methods are new, they have not been sufficiently tested in cases relevant for numerical simulations of clouds. The topic of the presented work is therefore very interesting and fits well into the scope of the GMD journal.

**General comments**

The presentation of the design of the three different collision algorithms is done very well. Figure 2, combined with the detailed description of collision algorithms and the "hypothetical algorithm", clearly shows the differences in treatment of collisions inherent to these three algorithms. Also, the presentation of the three different initialization procedures is done well.

The Authors did an immense job at testing different algorithm options and simulation parameters. The Authors have tested 3 different algorithms ("remapping" (RMA), "average impact" (AIM) and "all-or-nothing" (AON) algorithms), used 3 different test cases (Golovin, Long and Hall kernels), 3 different initialization procedures and many different collision algorithm options and simulation parameters. The Authors have tested a big parameter space and it is a big achievement of the presented work. However, the presentation of the results from this set of tests could be improved. In my opinion the big number of figures showing results from many combinations of simulation options makes the manuscript difficult to read and pinpoint the interesting and beneficial parameter combinations. Instead of providing a report from many test simulations that were made, a more concise summary of obtained results would be more beneficial and easier to comprehend for the reader, in my opinion. In general, the quality of many figures in the manuscript is poor and sometimes makes them

impossible for me to comprehend.

The final analysis of the accuracy of the three tested collision algorithms is very critical, witch is a good aspect of the manuscript. The Lagrangian schemes are free of many numerical limitations of the bin schemes, but they do introduce new numerical challenges that need to be addressed. The big number of tests performed by the Authors allows a detailed analysis of accuracy. The final discussion of the collision algorithms could also underline some advantages of the Lagrangian schemes.

**Overall, the manuscript discusses an interesting topic and provides a wide variety of tests. The corrections suggested in the following part of my review are minor and focus mostly on improving the figures.**

**Specific comments**

**1. "Too many" figures**

As stated before I think that the Authors did a tremendous job implementing the three algorithms and then testing them in such a large variety of simulations. Nevertheless, in my opinion, some parts of the presentation could be improved by removing figures and providing instead a summary of the obtained results. Below I'm including some suggestions on how it might be done.

For the sake of completeness and some potential future comparisons with other algorithms I would suggest moving some figures to electronic supplement. Such supplement could contain all figures, data needed to plot them and (if the Authors are willing) the scripts used for plotting. This would enable other Lagrangian scheme

[Figure]

users to test their own implementations and then easily plot their own results against the tests performed by the Authors. A good example of such electronic supplement is in the Lauritzen 2014 GMD paper (doi: $10.5194/\text{gmd}-7-105-2014$). Note, that such supplement does not demand publishing the actual code of the three algorithms but only simulation results. This is easier to do and to document.

List of figures that could be moved to supplement:

- **6, 9, 15:** Both AIM and AON algorithms do not change the number of SIPs ($N_{SIP}$). Maybe stating in the legend what was the number of SIPs used is enough and the first row of plots could be redundant. For the RMA algorithm it would be more beneficial for me to provide the actual number of the additional SIPs needed (for example as a % of the initial number of SIPs). For the size distribution moments it would be more beneficial for me to introduce some measure of error and then report the error value for different combinations of parameters that are tested. – In most of the plots the lines are on top of each other and are therefore not readable. A table of error values would be easier to read.

- **11, 19, 22 - top row and the last or second to last row:** Similar to the previous comment, the $N_{SIP}$ is constant and therefore a clear legend instead of the first row of plots would be enough in my opinion. The behavior of the second and the third moment is very similar and I think that one row of panels could be omitted. The behavior could be only described in text. Again, introducing some error measure and reporting its value would be more informative for me. It would help to summarize all the results and enhance the comparison between different options and algorithms.

- **13:** The behavior for Hall kernel is similar to Long kernel (Fig. 11) Perhaps stating that in text could be sufficient?

- **17:** Similar to Fig. 11 and 19, maybe just two size distribution moments are sufficient? Again, some error measure would be useful.

**2. Comments on figures**

All the figures presented in the manuscript are too small for me to read easily. Also the font size and the line thickness is too small.

The color-coding and plot styles of some of the figures make them difficult to read for me. Below I'm including a list of such figures with some ideas on how they could be improved:

- **2 – RMA algorithm:** The gray font color used for text regarding contribution k is not readable. Maybe just for text a darker color could be used?

- **3, 4 – top panel:** The number of points and the chaotic color-coding makes it impossible for me to easily see what is happening in the left part of the plot. Reducing the number of SIPs shown, especially for the small drop sizes, would help. I would also suggest choosing line colors basing on the initial drop size rather than at random – for example http://stackoverflow.com/questions/13972287/having-line-color-vary-with-data-index-for-line-graph-in-matplotlib

- **5, 7, 8, 10, 12, 14, 16, 18:** Similar to the previous case I would suggest choosing line colors basing on the simulation time rather than at random. Especially for later figures showing oscillations for RMA or less smooth solutions for AON it would make it easier to compare different lines.

- **11, 13, 19, 20, 22:** Similar to the previous case, consider choosing colors basing on the number of SIPs used. It would make the first row of plots unnecessary and allow easier comparison.

- **16:** In my opinion showing just one realisation and the average over 50 realisations could be enough. It's obvious that any realisation from AON will be burdened with irregular scatter. It's also obvious that averaging over even bigger ensemble will further smooth the solution and it could be just stated in text. Gained space could be then used to increase the size of the plots. The symbols *, + and - in the last two panels are not readable in a plot of this size and obscure the lines representing the actual size distribution.

- **17:** The red and green colors overlay each other and make it difficult to read the figure. I'd suggest omitting one size distribution moment and using the space to significantly increase the size of the plot as well as the size of points and line thickness.

**3. Pseudo-code listings**

Please consider providing an additional caption explaining the conventions used in the listing. What lines are marked as comments and what lines are the actual pseudo-code? What does it mean if a line is written in italics, bold or in capital letters?

**4. Discussion for Long kernel**

The bin scheme solves the Smoluchowski equation for the number concentration function and by default should provide a smooth solution. However, the Smoluchowski equation is strictly true for infinite systems. For cases of big population of similar drops (i.e. a population of rain drops from a fully formed precipitation event) solving the Smoluchowski equation provides a good representation of the drop size distribution. In contrast, the onset of precipitation (or the "transition phase" for the Long kernel in 30-40 minutes of simulation time) might be governed by the

behavior of just a few big "lucky" drops. See for instance the discussion in Lushnikov 2004 (doi: `10.1103/PhysRevLett.93.198302`) and Bayewitz et. al. 1974 (doi: `10.1175/1520-0469(1974)031<1604:TEOCIA>2.0.CO;2`) The bin solutions are commonly considered a true solutions during comparison studies. However it is not clear to me what volume should be used in order to ensure that solving the Smoluchowski equation is a good method for all precipitation phases. A discussion of issues related to this topic is definitely out of the scope of this manuscript. However, could you consider adding a small warning or comment on this aspect?

In the summary of box model tests, could you outline in text how the difficulties encountered in the transition phase of the Long kernel actually affect the final solution at t=60 for RMA, AIM and AON for the best combination of the algorithm options? Do the oscillations in RMA and scatter in AON preclude a good final solution? How accurate is the final stable and smooth solution from AIM in comparison? Is the location and value of the final maximum easily captured in AIM and AON?

**5. Other comments**

- line 221 - Could you comment on what techniques do you recommend when fighting numerical cancellation errors? What procedure was used in the current implementation?

- line 252-253 - Could you comment on why the described behavior is considered advantageous?

- line 273 - Maybe consider stating what initialization will be used as default in the later box model tests?

- Pseudocode for RMA, line 30 - is $N_{SIP} = ii$ or should it be $i$?

- Figure 3 and 4 bottom panel - normalizing once with regard to the initial condition and once with regard to the final state is confusing

- line 488 - Another alternative could be to assign the product of collision to just one SIP and use the remaining SIP to split the biggest weighting factor between two SIPs. See the third to last paragraph in sec. 5.4.1 in Arabas 2015

- line 535 - In my opinion performing collisions only for selected random pairs and scaling the probability is a very useful feature. It changes the asymptotic behavior of the scheme with regard to the number of SIPs from quadratic to linear. It allows to perform simulations with a bigger number of SIPs, which increases the resolution of the obtained results. Could you consider underlying those benefits? If some further tests are planned for the future, I would suggest adding this option to the AON implementation. On a side note, we use AON with collisions for random pairs and singleSIP init by default in our Lagrangian simulations. Out of curiosity, we ran the Long and Hall tests described in the manuscript using our default parameters. The results are similar to those presented by the Authors for AON box model tests.

- Figures 5, 6, 7 are not averaged over 50 realisations. In contrast, the corresponding figures for AIM and AON are. Could you comment on why? Does the design of RMA algorithm guarantee no need for ensemble runs? Could the ensemble runs be obtained using one of the random initialization procedures? For Golovin kernel RMA produces good results for a single realisation, which should be underlined. If for Long and Hall kernels ensemble average does not help, it should also be underlined. Could you comment on how an ensemble average for RMA for high SIP number (for example $\kappa = 200$) for Long kernel would look? In general it was unclear for me if RMA (i) becomes unstable and does not provide a solution for Long test or (ii) is stable but generates cumbersome oscillations and wrong final solution.

- Is it necessary to average over an ensemble for AIM?

- line 796 - For the sake of summarizing the box model simulations, could you discuss in text what was a minimum number of SIPs and a maximum timestep needed to obtain satisfactory results for the best combination of options for RMA, AIM and AON? Was the computational cost of all algorithms similar? Does it scale in the same way when increasing SIP number? Could you summarize in text how sensitive the three algorithms are to timestep?

- line 814 - For me, the total number of SIPs is a more intuitive parameter than $\kappa$. Could you also state what is the total number of SIPs for $\kappa = 20$ and $\kappa = 100$?

- line 838 - Could you comment on why the RMA is excluded in this part of the study? Are the oscillations as prominent as in the Long test scenario? Does it again fail to reproduce the bin model results at the final stage?

- line 895 - Since the Authors state in line 858 that it is not clear which findings of the performed tests are most relevant for simulations of clouds, I would suggest somewhat weakening the statements about the RMA algorithm in the conclusions.

**Technical corrections**

- line 142 - k! should be the factorial not faculty?

- caption of Fig. 1 - $\lambda_3$ is missing

- Pseudocode for RMA, line 34 - should be "can be easily incorporated in ..."

- line 435 - should be Figure 3?

- line 560 - space missing after "per construction".

- Figure 14 is missing the third column that should depict a version of AON without self collections.

- line 834 - Could you rephrase the part "where the abundance of droplets larger than 10 um drops strongly"

---

## Author Comment (AC1) · 3 Mar 2017

*We want to thank reviewer 1 for his/her careful reading of the manuscript, the constructive and valuable comments. This document contains point-to-point replies to each point the reviewer made. The reviewer's comments are written in normal, our replies in bold italic font. Line numbers preceded by an exclamation mark refer to the original manuscript, those without to the revised one.*

The authors compare three different Langrangian Cloud Model (LCM) implementations with a focus on collection using three different collection kernels. Analytical solutions as well as previous bin model results are used as references. Additionally, sensitivity of the LCM implementations with respect to the initialization of the simulation particles (SIPs) as well as to different numerical features (resolution, time step, ...) is tested.

This results in a large amount of model runs with a great variety of possible parameter and configuration combinations which sometimes makes the manuscript difficult to read.

General comments

Each of the LCM implementations shows rather strong shortcomings:

• RMA cannot deal with realistic kernels (Long, Hall) and shows spurious oscillations.

• AIM systematically underestimates the collisional growth.

• AON always needs an ensemble of at least 50 realizations to reach a representative average result for the final drop size distribution since individual realizations deviate considerably from the average (in contrast to RMA and AIM). This severely limits the potential to be used in 2- and 3-dimensional models with a large number of grid points.

*We agree with the reviewer's opinion on RMA and AIM. For AON we are not as pessimistic as the reviewer regarding its use in 2-D and 3-D models. It will not be necessary to run 50 realisations in each grid box. In such cases, the averaging will occur over grid boxes with similar atmospheric conditions. We made a similar experience at least in simulations of contrail-cirrus, where we tested the NSIP-sensitivity of the deposition/sublimation process (see section 3.1 in Unterstrasser & Sölch, 2014). We found that very few SIPs per grid sufficed to reach convergence even though the few SIPs in a single grid box could not realistically represent a smooth size distribution. Smooth size distributions can be derived only for larger volumes of air when more SIPs are taken into account. This explanation is now given in section 4.1.*

*It will be interesting to see how AON performs in 2D/3D-setups. This will be the next step in an upcoming study.*

Additionally, sensitivities with respect to initialization of SIPs are shown to be high at least for some configurations. This problem is discussed towards the end of the manuscript where also more mature drop size distributions are used for initializations.

Within a full microphysics description including drop nucleation and condensational growth, it should be harder to control the DSD at the moment when collisions become important. This discussion

*We agree and it will be interesting to see how the algorithms behave in such a setup and what extensions may be necessary to guarantee optimal SIP weights. SIP splitting and merging as discussed in Unterstrasser & Sölch, 2014 may be an option.*

should be extended. Compared to spectral bin models the accuracy of all LCM implementations shown seems to be lower with at least comparable computational costs. One could conclude that LCMs are of no practical use. Nevertheless, LCMs are valuable tools. Please discuss critically advantages and disadvantages of LCMs.

*In the Introduction we add a paragraph mentioning the most important advantages/disadvantages of LCMs: "Due to their specific construction, LCMs offer a variety of advantages in comparison to spectral-bin and bulk cloud models. Their representation of aerosol activation and subsequent diffusional growth follows closely fundamental equations and avoids therefore the possible perils of parameterizations (e.g. Andrejczuk et al. 2008, Hoffmann 2016). The same applies for the representation of collection or aggregation, which is based on the interaction of individual SIPs. Accordingly, LCMs approximate pure stochastic growth (e.g. Gillespie 1975), which is the correct description of collection/aggregation within a limited system of interacting particles and results in the SCE, which is used as the basis for spectral-bin and bulk models, if the system becomes infinite (e.g. Bayewitz et al. 1974). Moreover, LCMs do not apply the finite-differences method to compute microphysics. Accordingly, LCMs are not prone to numerical diffusion and dispersion, and do not suffer from the numerical broadening of a droplet spectrum, which can affect spectral-bin cloud models (Khain et al.  2000). Finally, LCMs enable new ways of analysis by the tracking of individual SIPs. They can be used to reveal the origins of droplets, as well as conditions associated with their growth (e.g. Hoffmann et al. 2015, Naumann and Seifert 2016). The largest disadvantage of LCMs, so far, might be their relative novelty due to their higher computational demand. Many aspects of this approach have not been validated adequately or can be improved. For the process of collection/aggregation, this study will offer a first rigorous evaluation of the available numerical approaches. "*

*It is clear, that the present study is a first step of evaluation. Our next step is the AON evaluation in higher-dimensional tests. For this reason, we do not want to speculate too much about the performance in such applications. The initialisation of SIPs differs from model to model and/or application to application. Constant weights approaches, as used in several studies recently, greatly deteriorate the collection treatment. Those studies initialise the SIPs in the beginning of the simulation as CCN. Contrary to this, the LCM by Sölch and Kärcher, 2011, initialises SIPs only during cloud formation with varying SIP weights.*

*Regarding the computational costs no final conclusion can be made at the present stage. At least for localised cloud objects like contrails as studied with EULAG-LCM, the computational costs of LCMs and bin models are comparable. A bin model would carry out the same amount of computations in any grid box, whereas in LCM approaches no computations are carried out in the ice-free grid boxes (which can be the majority). On the other hand, this might not necessarily result in a model speed-up due to load-imbalancing.*

The quality of some figures is poor. Most of them are too small, lines are too thin and sometimes too many. Specific comments are given below.

**All figures have been revised.**

Specific comments

l. 154-157: What is the reason to switch from mass doubling (which is often used) to a tenfold increase as a basis for bin resolution?

*There is no particular reason besides personal flavour. A simple conversion formula is given in l. !157.*

l. 238: If the probabilistic version of the singleSIP-init is used, dots are not distributed uniformly!

*Strictly speaking, the dots are uniformly distributed only for the deterministic version. We hope that saying "homogeneously distributed" is an acceptable expression.*

Fig. 1: upper left and below: difference between red and green lines is misleading since the higher density of dots wrt the x-axis is not resolved.

*In the updated figure, κ is reduced such that the dots appear less connected.*

upper right: threshold radius line barely can be seen; lines for alpha-values are also misleading, can be confused with legends. Values should read N $10\alpha$ .

*Thank you for spotting the inconsistency with the usage of $\alpha$. We improved the figure.*

Last but one row: Is there a systematic difference between the symbols and the lines due to plotting issues? If not, a better initial agreement should be reached. Cp. also l. 268-270.

**The reason for the systematic difference between the symbols and the lines is the following: The data of the reference solution use a finer bin grid. Now we use the same κ as for the plots with the symbols and the agreement is better. As noted in the manuscript, we use $\kappa_{plot}$=4.**

Algorithm 1: What do k++ and i++ stand for? loops over k/i?

*This is C++ style for "k = k +1". We replace those expressions.*

Algorithm 2: l. 13: Please exchange gain and loss term due to consistency with l. 12 and eq. (22)

*Thanks for this hint. Corrected.*

Fig. 3: Top: It is difficult to see what happens in the left part of the spectrum (<20 μm). Bottom: It is confusing to normalize one ratio to t=0 and the other one to t=3600. Please redo the black curve with vi (t = 3600)/vi(t = 0)

*The motivation behind the definitions was that both fractions are >= 1 and both curves fit better in the same plot. We changed the description of the plot around line 473 to eliminate this pitfalll.*

l. 559 and Figs. 6, 9, 11, 13, 15, 17, 19, 22: The third moment λ3 should not be shown in the figures since the behaviour is very similar to λ2 (which should be stated in the text). The space saved can be used to extend some other figures in order to improve their readability.

*We remove all panels that show the third moments.*

Fig. 5: Use full lines with enhanced line thickness for RMA results and dotted (or dashed) lines for analytical solutions. Otherwise, all plots look identical at first glance.

**We changed it in all plots showing DSDs. Now the solid lines show our results, the dotted lines show the reference solution.**

l. 564-575: The discussion of the RMA Golovin results is very short and misses several aspects, e.g.: Why are the results for RedLim worse than the regular ones? What are the reasons for the relatively large differences between the two OTF versions?

*We expanded the description of the results in section 3.1.*

Fig. 6: Are there any lines missing? Variation of η only for κ = 60 in the left and the middle column? No κ = 60 for OTF at all? Which lines fall together and which runs are carried out at all?

*We added the necessary information in the caption to avoid any confusion. The variation of η was indeed only shown for κ =60. And in the right panel only four simulations were shown. We redesigned the plot and the selection of simulation at display changed slightly.*

Fig. 7: see Fig. 5: full lines for the RMA results.

**Done.**

l. 595: Compared to the regular version (and to bin model results) I would not call the RMA RedLim results "perfect". The same holds for the OTFs results; only OTFl is almost "perfect".

**We agree and reformulated the paragraph in section 3.1.**

Figs. 8, 10, and 12: see Fig. 5: full lines for the AIM results.

**Done.**

Fig. 14: see Fig. 5: full lines for the AON results. Results plot for disregarding self-collections is missing.

*Good point. We decided at some point to leave out the third panel and forgot to change the caption.*

l. 739: Is this restricted to AON results or do the other methods show similar robustness wrt to the small tail of the distribution? In reality very small drops similarily do not contribute substantially to the growth of the large mode due to their low number and small individual mass. This should be reflected in model sensitivities.

**This insensitivity to the small tail is even more pronounced in the AIM algorithm. This was already mentioned in the original manuscript around line !642.**

l. 746: This is in contrast at least to the Golovin RMA results. Why is it reasonable for AON? Is this due to the lower number of collision events realised because of the probability restrictions?

*Yes, this is the reason. We added an explanation around line 815.*

l. 766: When pcrit is smaller, less collection events can be expected (see lines 469/470). A spread in v-values leads to smaller and larger v-values. Does this mean that the largest v-values are responsible for the enhanced collection?

**The sentence contains a typo. "$p_{crit}$ is smaller" must be replaced by "$p_{crit}$ is larger". A combination of a small v SIP and a large v SIP leads to a large $p_{crit}$.**

l. 902ff: It should be critically mentioned that AON always needs an ensemble of at least 50 realizations to reach a representative average result for the final drop size distribution since individual realizations deviate considerably from the average (in contrast to RMA and AIM). This leads to a large effort in terms of computational resources.

*See our response in the beginning of this reply. We included our thoughts in section 4.1.*

Technical corrections

***Thank you for spotting all those typos and inconsistencies. They have been corrected.***

l. 18: ... are important processes ...

Table 1: mean mass: M/N

Fig. 1 caption: alpha should be (-2, -3, -7)

l. 256: values of N $10\alpha$ ...

l. 276: However, it is ...

l. 371: rather "proposed" than "discussed"

l. 410ff.: The terms "larger SIP" and "smaller SIP" are used here with the meaning "SIP with larger/smaller drops(=higher average drop mass)". Please define whether "large SIP" indicates large drops or a large number of drops within the SIP (cp. l. 510).

***We intended to use large/small SIP throughout the text in the sense you mention. We added a sentence on the terminology around line 120.***

Fig. 3 caption: ... function of their initial radius ... Please add that it is an AIM simulation.

l. 434: ... of each droplet within the SIP ...

l. 435: Figure 3

l. 510: In contrast to l. 410ff. smallest SIP refers to the size of the droplets not the weighting factors;

***The description is correct the way it is. In both paragraphs small SIP refers to a SIP with small droplets and the terminology is consistently applied.***

Fig. 11 caption: ... black curves with triangles ... green lines for vrandom; $\alpha$ should be (-2, -3, -7)

l. 658: green lines

l. 792: check the meaning of "large SIP", also l. 824 "heavy SIP"

***Again large SIP refers to a large single droplet mass. Heavy SIP are SIPs with large total mass $\chi = v \mu$.***

---

## Author Comment (AC2) · 3 Mar 2017

We want to thank Anna Jaruga for her careful reading of the manuscript and her constructive and valuable comments. This document contains point-to-point replies to each point the reviewer made. The reviewer's comments are written in normal, our replies in bold italic font. Line numbers preceded by an exclamation mark refer to the original manuscript, those without to the revised one.

The Authors evaluate three algorithms for representing collisions in Lagrangian cloud microphysics schemes that are available in the literature. The design and some implementation details are discussed. The accuracy of the three collision algorithms is compared against the analytical solution of Golovin kernel and bin solutions of Long and Hall kernels. A very wide parameter space is investigated. The manuscript also tests three different initialization techniques for the Lagrangian cloud microphysics schemes.

The work presented here is very useful. Lagrangian schemes offer very detailed representation of microphysical processes in clouds. Yet, because the Lagrangian methods are new, they have not been sufficiently tested in cases relevant for numerical simulations of clouds. The topic of the presented work is therefore very interesting and fits well into the scope of the GMD journal.

**General comments**

The presentation of the design of the three different collision algorithms is done very well. Figure 2, combined with the detailed description of collision algorithms and the "hypothetical algorithm", clearly shows the differences in treatment of collisions inherent to these three algorithms. Also, the presentation of the three different initialization procedures is done well. The Authors did an immense job at testing different algorithm options and simulation parameters. The Authors have tested 3 different algorithms ("remapping" (RMA), "average impact" (AIM) and "all-or-nothing" (AON) algorithms), used 3 different test cases (Golovin, Long and Hall kernels), 3 different initialization procedures and many different collision algorithm options and simulation parameters. The Authors have tested a big parameter space and it is a big achievement of the presented work. However, the presentation of the results from this set of tests could be improved. In my opinion the big number of figures showing results from many combinations of simulation options makes the manuscript difficult to read and pinpoint the interesting and beneficial parameter combinations. Instead of providing a report from many test simulations that were made, a more concise summary of obtained results would be more beneficial and easier to comprehend for the reader, in my opinion. In general, the quality of many figures in the manuscript is poor and sometimes makes them. The final analysis of the accuracy of the three tested collision algorithms is very critical, witch is a good aspect of the manuscript. The Lagrangian schemes are free of many numerical limitations of the bin schemes, but they do introduce new numerical challenges that need to be addressed. The big number of tests performed by the Authors allows a detailed analysis of accuracy. The final discussion of the collision algorithms could also underline some advantages of the Lagrangian schemes.

Overall, the manuscript discusses an interesting topic and provides a wide variety of tests. The corrections suggested in the following part of my review are minor and focus mostly on improving the figures.

**Specific comments**

**1. "Too many" figures**

As stated before I think that the Authors did a tremendous job implementing the three algorithms and then testing them in such a large variety of simulations. Nevertheless, in my opinion, some parts of the presentation could be improved by removing figures and providing instead a summary of the obtained results.

**We agree that there are many figures.**

Below I'm including some suggestions on how it might be done. For the sake of completeness and some potential future comparisons with other algorithms I would suggest moving some figures to electronic supplement. Such supplement could contain all figures, data needed to plot them and (if the Authors are willing) the scripts used for plotting. This would enable other

We created a supplement that contains a systematic and comprehensive collection of figures (showing around 100 sensitivity tests). The standardized figures show the temporal evolution of NSIP and the moments 0, 2 and 3 (analogous to the plots shown in the original manuscript version). This collection discloses in detail the behaviour of the three algorithms, will facilitate future comparisons by fellow LCM developers and guide them. The underlying data (150 RMA, 270 AIM and 400 AON simulations) are not included in the supplement. For many simulations this makes no sense, as the algorithms obviously do not produce optimal results or the results are similar to basically identical. Our data can be obtained by request. Regarding the well-established Bott and/or Wang solutions, both scientists are happy sharing their data and we recommend addressing them directly.

Lagrangian scheme users to test their own implementations and then easily plot their own results against the tests performed by the Authors. A good example of such electronic supplement is in the Lauritzen 2014 GMD paper (doi: 10.5194/gmd-7-105-2014). Note, that such supplement does not demand publishing the actual code of the three algorithms but only simulation results. This is easier to do and to document.

List of figures that could be moved to supplement:

• 6, 9, 15: Both AIM and AON algorithms do not change the number of SIPs (NSIP). Maybe stating in the legend what was the number of SIPs used is enough and the first row of plots could be redundant. For the RMA algorithm it would be more beneficial for me to provide the actual number of the additional SIPs needed (for example as a % of the initial number of SIPs). For the size distribution moments it would be more beneficial for me to introduce some measure of error and then report the error value for different combinations of parameters that are tested. – In most of the plots the lines are on top of each other and are therefore not readable. A table of error values would be easier to read.

Our results have shown that the representation of collection within a DSD depends significantly on the initial DSD, its representation by SIPs (mass and weighting factor), the number of SIPs itself, the simulated time, as well as the applied collection kernel. Due to this strong case dependency, a quantitative error measure might favour the misuse of our results if there is no comparability. (Example: I used 200 SIPs and therefore the error due to collection should be 5 %.) Therefore, we would like to remain at our rather qualitative error analysis, which, however, will serve as a basis for decisions on numerical parameters for future studies. (Example: The more SIPs the better the representation.)

• 11, 19, 22 - top row and the last or second to last row: Similar to the previous comment, the NSIP is constant and therefore a clear legend instead of the first row of plots would be enough in my opinion. The behavior of the second and the third moment is very similar and I think that one row of panels could be omitted. The behavior could be only described in text. Again, introducing some error measure and reporting its value would be more informative for me. It would help to summarize all the results and enhance the comparison between different options and algorithms.

Our intention was to give to a complete overview of our results such that other developers of LCMs can best benefit from our experience. Based on both reviews, we realise that a better pre-selection of results that are shown is necessary. Hence, we follow your advices. We removed all rows showing the third moment and the SIP number evolution in AIM and AON and increase the remaining panels. We inserted a table with the  $N_{SIP}$ -values.

**And the results are summarized in the newly introduced section 4.1**

• 13: The behavior for Hall kernel is similar to Long kernel (Fig. 11) Perhaps stating that in text could be sufficient?

**All figures with Hall simulations are moved to the supplement.**

• 17: Similar to Fig. 11 and 19, maybe just two size distribution moments are sufficient? Again, some error measure would be useful.

**Done.**

**2. Comments on figures**

All the figures presented in the manuscript are too small for me to read easily. Also the font size and the line thickness is too small.

The color-coding and plot styles of some of the figures make them difficult to readme. Below I'm including a list of such figures with some ideas on how they could be improved:

• 2 – RMA algorithm: The gray font color used for text regarding contribution k is not readable. Maybe just for text a darker color could be used?

**A darker color is used now and font size has been increased where possible.**

• 3, 4 – top panel: The number of points and the chaotic color-coding makes it impossible for me to easily see what is happening in the left part of the plot. Reducing the number of SIPs shown, especially for the small drop sizes, would help. I would also suggest choosing line colors basing on the initial drop size rather than at random – for example

http://stackoverflow.com/questions/13972287/having-line-color-vary-with-data-index-for-linegraph-in-matplotlib

I read the suggested web page. It is definitely worth to think about the colour coding. I personally favour colours, which I can assign a name ('red', 'blue'). For the suggested colour ranges, it is hard to use unambiguous colour names in the text. Figure 3 and 4 top are improved by showing fewer SIPs and we keep the original colours.

• 5, 7, 8, 10, 12, 14, 16, 18: Similar to the previous case I would suggest choosing line colors basing on the simulation time rather than at random. Especially for later figures showing oscillations for RMA or less smooth solutions for AON it would make it easier to compare different lines.

**Increasing the thickness of the curves and using solid lines for the simulation results (the reference is now plotted with dotted lines) hopefully makes the plots better readable. Any of the plots you mention contains a legend with the time. So we think it is o.k. to leave the colours as they are.**

• 11, 13, 19, 20, 22: Similar to the previous case, consider choosing colors basing on the number of SIPs used. It would make the first row of plots unnecessary and allow easier comparison.

**The rows showing $N_{SIP}$ have been removed from the plots.**

• 16: In my opinion showing just one realisation and the average over 50 realisations could be enough. It's obvious that any realisation from AON will be burdened with irregular scatter. It's also obvious that averaging over even bigger ensemble will further smooth the solution and it could be just stated in text. Gained space could be then used to increase the size of the plots. The symbols \*, + and – in the last two panels are not readable in a plot of this size and obscure the lines representing the actual size distribution.

**We use a new layout. The interquartile range is now shown in a separate plot. To illustrate the probabilistic nature of AON we think it is justified to show two realisations.**

• 17: The red and green colors overlay each other and make it difficult to read the figure. I'd suggest omitting one size distribution moment and using the space to significantly increase the size of the plot as well as the size of points and line thickness.

**We replotted the figure and hope it is better readable now.**

**3. Pseudo-code listings**

Please consider providing an additional caption explaining the conventions used in the listing. What lines are marked as comments and what lines are the actual pseudo-code? What does it mean if a line is written in italics, bold or in capital letters?

**A paragraph on layout conventions is added at the end of section 2.2.**

**4. Discussion for Long kernel**

The bin scheme solves the Smoluchowski equation for the number concentration function and by default should provide a smooth solution. However, the Smoluchowski equation is strictly true for infinite systems. For cases of big population of similar drops (i.e. a population of rain drops from a fully formed precipitation event) solving the Smoluchowski equation provides a good representation of the drop size distribution. In contrast, the onset of precipitation (or the "transition phase" for the Long kernel in 30-40 minutes of simulation time) might be governed by the behavior of just a few big "lucky" drops. See for instance the discussion in Lushnikov 2004 (doi:

10.1103/PhysRevLett.93.198302) and Bayewitz et. al. 1974 (doi: 10.1175/1520-

0469(1974)031<1604:TEOCIA>2.0.CO;2) The bin solutions are commonly considered a true solutions during comparison studies. However it is not clear to me what volume should be used in order to ensure that solving the Smoluchowski equation is a good method for all precipitation phases. A

discussion of issues related to this topic is definitely out of the scope of this manuscript. However, could you consider adding a small warning or comment on this aspect?

We added a small comment in the introduction(around line 87) that the LCM rather solves the pure stochastic growth (Gillespie 1975) than the SCE due to the consideration of individual collisions within a finite system of particles. However, the pure stochastic growth approaches the SCE for  $N \rightarrow \infty$  or, equivalently, as the number of realizations approaches infinity, which can be seen from Fig. 15.

In the summary of box model tests, could you outline in text how the difficulties encountered in the transition phase of the Long kernel actually affect the final solution at t=60 for RMA, AIM and AON for the best combination of the algorithm options?

Using the "best combination" of options, the reference solution of the spectral bin model was relatively well captured by any algorithm (e.g., by comparing the second moment for the RMA, AIM, and AON algorithms, which can be interpreted as a proxy of the largest droplets here). However, the "best combination" might not be the best combination for application in a higher dimensional model. As outlined in the Discussion and Conclusion section of our manuscript, RMA might depend on an infeasibly small time step, AIM might suffer from an inappropriate initial distribution of weighting factors. Only AON has been shown to handle most "situations" successfully. Therefore, we propose the AON algorithm for practical purposes.

Do the oscillations in RMA and scatter in AON preclude a good final solution? How accurate is the final stable and smooth solution from AIM in comparison? Is the location and value of the final maximum easily captured in AIM and AON?

These points are covered in the Conclusions of our manuscript. For RMA, the oscillations indeed preclude a good final solution if a feasible time step is used. To overcome the scatter in AON, a sufficient number of realisations is needed. And the AIM solution is stable and smooth but inherently depends on the initial distribution of SIPs.

**5. Other comments**

• line 221 - Could you comment on what techniques do you recommend when fighting numerical cancellation errors? What procedure was used in the current implementation?

We rephrased the paragraph. Numerical cancellation errors are smaller when expressions can be reformulated such that differences of similarly valued terms are eliminated. More on this topic can be found in classical textbooks on numerical analysis.

• line 252-253 - Could you comment on why the described behavior is considered advantageous?

**We reformulated the sentence. See line 273.**

• line 273 - Maybe consider stating what initialization will be used as default in the later box model tests?

**Done. See line 295.**

• Pseudocode for RMA, line 30 - is NSIP = ii or should it be i?

**Thanks, corrected.**

• Figure 3 and 4 bottom panel - normalizing once with regard to the initial condition and once with regard to the final state is confusing

**Reviewer 1 had the same comment and we changed the description to use analogous definitions for both quantities (around line 473).**

• line 488 - Another alternative could be to assign the product of collision to just one SIP and use the remaining SIP to split the biggest weighting factor between two SIPs. See the third to last paragraph in sec. 5.4.1 in Arabas 2015.

You refer to section 5.1.4, not 5.4.1. Yes this is also an option. It is relevant for approaches where you want to keep the total SIP number constant. Otherwise, you could simply delete one of the two SIPs and introduce a SIP splitting independently of whether it just happened that you save one SIP during an equal-v collection.

• line 535 - In my opinion performing collisions only for selected random pairs and scaling the probability is a very useful feature. It changes the asymptotic behaviour of the scheme with regard to the number of SIPs from quadratic to linear. It allows to perform simulations with a bigger number of SIPs, which increases the resolution of the obtained results. Could you consider underlying those benefits?

If some further tests are planned for the future, I would suggest adding this option to the AON implementation. On a side note, we use AON with collisions for random pairs and singleSIP init by default in our Lagrangian simulations. Out of curiosity, we ran the Long and Hall tests described in the manuscript using our default parameters. The results are similar to those presented by the Authors for AON box model tests.

It is nice to see that the manuscript was an incentive for you to carry out test runs. The computational cost of the current implementation is quadratic in terms of  $N_{SIP}$  per grid box (clearly not the total  $N_{SIP}$  of all grid boxes). Clearly, linear costs sound much better. But it is clear that this describes the asymptotic behaviour and becomes important for large  $N_{SIP}$ . My impression is that linear sampling was introduced because in the initial work of Shima tremendously many SIP were used and simulations weren't feasible at all with quadratic costs. The question will be if you need more SIPs to reach convergence when you use a linear sampling. In a full microphysical model this  $N_{SIP}$ -increase affects the costs of all the other processes as well. Nevertheless, in more complex settings the linear sampling is definitely a viable option which deserves to be tested. See paragraph in new section 4.1.

• Figures 5, 6, 7 are not averaged over 50 realisations. In contrast, the corresponding figures for AIM and AON are. Could you comment on why? Does the design of RMA algorithm guarantee no need for ensemble runs? Could the ensemble runs be obtained using one of the random initialization procedures? For Golovin kernel RMA produces good results for a single realisation, which should be underlined. If for Long and Hall kernels ensemble average does not help, it should also be underlined. Could you comment on how an ensemble average for RMA for high SIP number (for example  $\kappa = 200$ ) for Long kernel would look? In general it was unclear for me if RMA (i) becomes unstable and does not provide a solution for Long test or (ii) is stable but generates cumbersome oscillations and wrong final solution.

Originally, we didn't actually run the RMA program for a full ensemble with 50 realisations as it was done for the other algorithms. This has two different reasons:

1. As we struggled to get realistic Long/Hall solutions with RMA, we did many tests with program versions that computed only one realisation. We never extended our implementation beyond this test framework.

2. After each time step of RMA, a new list of SIPs is created. From each bin, one SIP is generated using the exact bin values (see line 28 in Algorithm). There is no probabilistic component included in the algorithm.

To be more consistent in the presentation of the results, we programmed a version for multiple realisations and now show averages for RMA as we already did for AIM and AON. Figures with the RMA  $\lambda$ -evolutions, which have not been shown in the original submission, are now included in Fig.8. We changed the description of the text accordingly.

Concerning your last question: It is stable in the sense that the total mass is conserved, the higher moments do not explode and one can perform simulations over the whole 60 minutes. Indeed, Figure 7 shows size distributions for t=60min. If you call this behaviour unstable or not, depends also on how stability is defined in this context. In the end, what matters is that RMA solutions are far from the reference solution and useless (at least for feasible time steps). Before we started our evaluations, we expected that RMA is the best algorithm as we thought that the remapping is a clever approach. Hence we tested many options trying to get it "stable", but we did not succeed in this. We make this point hopefully a bit clearer by the extended Figure 7 and the added Figure 8. We also no longer use the word "stable" in the text now.

• Is it necessary to average over an ensemble for AIM?

**One AIM simulation is enough. It is not necessary to average over an ensemble, as mentioned in *I.!665-667*. In the revised version, we repeat this finding in section 4.1.**

• line 796 - For the sake of summarizing the box model simulations, could you discuss in text what was a minimum number of SIPs and a maximum timestep needed to obtain satisfactory results for the best combination of options for RMA, AIM and AON? Was the computational cost of all algorithms similar? Does it scale in the same way when increasing SIP number? Could you summarize in text how sensitive the three algorithms are to timestep?

**We added a paragraph on time step sensitivity of all three algorithms and on parameter requirements for convergence in section 4.1.**

• line 814 - For me, the total number of SIPs is a more intuitive parameter than kappa. Could you also state what is the total number of SIPs for  $\kappa = 20$  and  $\kappa = 100$ ?

We rephrased the paragraph (around line 935) to add the information on  $N_{\text{SIP}}$ .

Nonetheless, a small side remark:  $N_{SIP}$  is not a parameter of the singleSIP-init, it is diagnosed. This is why we stick to  $\kappa$  when we talk about the singleSIP-init. For  $v_{draw}$  and  $v_{const}$ -methods, the story is different as  $N_{SIP}$  is directly prescribed.

• line 838 - Could you comment on why the RMA is excluded in this part of the study? Are the oscillations as prominent as in the Long test scenario? Does it again fail to reproduce the bin model results at the final stage?

**Good point. We carried out RMA simulation with the later initialisations. RMA fails again. The simulations are shown in the supplement.**

• line 895 - Since the Authors state in line 858 that it is not clear which findings of the performed tests are most relevant for simulations of clouds, I would suggest somewhat weakening the statements about the RMA algorithm in the conclusions.

**Good point.**

**Technical corrections**

- line 142 k! should be the factorial not faculty?
- caption of Fig. 1  $\lambda$ 3 is missing
- Pseudocode for RMA, line 34 should be "can be easily incorporated in ..."
- line 435 should be Figure 3?
- line 560 space missing after "per construction".
- Figure 14 is missing the third column that should depict a version of AON without self collections.

**Good point. We decided at some point to leave out the third panel and forgot to change the caption.**

• line 834 - Could you rephrase the part "where the abundance of droplets larger than 10 um drops strongly"

**All technical corrections have been done.**